# Obese mother offspring have hepatic lipidic modulation that contributes to sex-dependent metabolic adaptation later in life

Christina Savva[1,2], Luisa A. Helguero [3], Marcela González-Granillo[1,2], Daniela Couto [4,5], Tânia Melo[4,5], Xidan Li[1], Bo Angelin [1,2], Maria Rosário Domingues[4,5], Claudia Kutter [6] & Marion Korach-André [1,2✉]

With the increasing prevalence of obesity in women of reproductive age, there is an urgent need to understand the metabolic impact on the fetus. Sex-related susceptibility to liver diseases has been demonstrated but the underlying mechanism remains unclear. Here we report that maternal obesity impacts lipid metabolism differently in female and male offspring. Males, but not females, gained more weight and had impaired insulin sensitivity when born from obese mothers compared to control. Although lipid mass was similar in the livers of female and male offspring, sex-specific modifications in the composition of fatty acids, triglycerides and phospholipids was observed. These overall changes could be linked to sex-specific regulation of genes controlling metabolic pathways. Our findings revised the current assumption that sex-dependent susceptibility to metabolic disorders is caused by sex-specific postnatal regulation and instead we provide molecular evidence supporting in utero metabolic adaptations in the offspring of obese mothers.

[1] Department of Medicine, Cardio Metabolic Unit (CMU) and KI/AZ Integrated Cardio Metabolic Center (ICMC), Karolinska Institute at Karolinska University Hospital Huddinge, Stockholm, Sweden. [2] Clinical Department of Endocrinology, Metabolism and Diabetes, Karolinska University Hospital Huddinge, Stockholm, Sweden. [3] Institute of Biomedicine, Department of Medical Sciences, University of Aveiro, Aveiro, Portugal. [4] CESAM, Centre for Environmental and Marine Studies, Department of Chemistry, University of Aveiro, Santiago University Campus, Aveiro, Portugal. [5] Mass Spectrometry Centre, LAQV-REQUIMTE, Department of Chemistry, University of Aveiro, Santiago University Campus, Aveiro, Portugal. [6] Department of Microbiology, Tumor and Cell Biology, Science for Life Laboratory, Karolinska Institute, Stockholm, Sweden. ✉email: marion.korach-andre@ki.se

Numerous experimental animal studies as well as epidemiological observations have demonstrated that exposure to environmental factors, including diet, contributes to chronic diseases. Surprisingly, risks for pathological disorders can be established in utero, often in the first trimester of pregnancy, and will affect the health of the individual later in life despite healthy postnatal lifestyle. The increased prevalence of overweight and obesity in women of reproductive age in both low-to-middle and high-income countries has been associated with increased obesity and metabolic diseases in their children[1–3]. Animal and human studies have shown that high-calorie diet intake by mothers during premating, gestation, and lactation has a permanent impact on the offspring's body composition, cardio-, and metabolic health[2,4,5]. Maternal obesity (MO) has been linked to insulin resistance (IR), glucose intolerance, and increased fat mass in offspring[4,6] due to epigenetic modifications that prime gene regulatory responses in the offspring[7].

Sex-dependent differences in body fat distribution and composition are key factors which are associated with pathophysiological consequences, such as low-grade inflammation, liver diseases, and IR. Indeed, visceral fat content has been correlated with metabolic dysfunctions, including liver steatosis[8], while subcutaneous fat accumulation has been shown to be protective[9]. Dysregulated lipid metabolism is an established risk factor for the development of liver steatosis and IR since molecular lipid species act by regulating insulin signaling and inflammatory response either positively or negatively in several organs including the liver[10]. Recently, distinct sex-specific lipid species have been identified that are key components in the development of metabolic dysfunctions in obesity[11,12]. The mechanism causing sex-specific metabolic differences involves a complex control of gene regulatory programs. Studies in mice showed that MO gives rise to distinct gene expression signatures in the livers of male and female offspring[4–6,13]. Closer inspection revealed an estrogen-dependent regulation of key genes in lipid pathways, which may explain the sex-specific metabolic responses in the offspring later in life[5,11,14,15].

Even though recent studies have shown intriguing sex differences in the metabolic response to obesity, both in human[16] and rodent models[11,17,18], sex-dependent adaptation to high-calorie intake is still unclear. In this study, we combined controlled mouse experimentation with state-of-the-art non-invasive magnetic resonance techniques and lipidomic analysis to monitor hepatic lipid species at different developmental time points in life. MO induced a panoply of events that sculpted the hepatic lipidome in a sex-dependent manner. Our findings show that metabolic adaptation to MO in the offspring is sex-dependent due to sex-specific transcriptional and post-transcriptional regulation of genes involved in lipid metabolism. Although primarily addressing the effects in liver, these consequences may compromise the functions of other organs.

## Results

### Physiological adaptation to MO in offspring is sex-dependent.
Mothers (F0 dam) received either the control diet (CD) or the high-fat diet (HFD) for 6 weeks before mating, during pregnancy, and lactation. F0 sires got the CD throughout the study. All F1 offspring were fed the CD after weaning. We refer to offspring born from HFD mothers as HF/C and those born from CD mothers as C/C (Fig. 1a). Mothers on HFD gained significantly more weight before mating than mothers on CD (Supplementary Fig. S1a). Irrespective of the maternal diet, females (F) weighed less than males (M) from week 8 of age until sacrifice. In contrast to F, HF/C M gained more weight than C/C M (Fig. 1b). The average food intake during the postnatal period was higher in M than in F regardless of the maternal diet (Supplementary Fig. S1b). These data demonstrate that MO impacts the body weight (BW) of M but not F offspring without changing food consumption behavior.

Given the higher BW in HF/C M compared to C/C M, we tested for changes in total body fat (TF) content at 15 weeks (mid term, MID) and 25 weeks (end term, END) of age by magnetic resonance imaging (MRI). MRI confirmed that HF/C M but not HF/C F accumulated more fat than C/C at MID, which was normalized at END (Fig. 1c, d). Closer inspection of subcutaneous (SAT) and visceral (VAT) adipose tissue showed sex difference in fat distribution. SAT and VAT appeared unchanged in F; however, HF/C M showed reduced SAT at MID and at END but gradually more VAT than C/C M and F groups (Fig. 1e, f). In sum, our results reveal that MO affects fat distribution in offspring in a sex-specific manner and that post-weaning CD of offspring can diminish the effects of MO on body fat in the long term (25 weeks) in both sexes. However, M offspring showed a redistribution of body fat toward more VAT along with less SAT, which is correlated with metabolic dysfunctions.

### Male offspring born from obese mothers show metabolic dysfunctions.
Adipose tissue remodeling is an established risk factor for the development of IR. To assess metabolic parameters indicative for IR, we measured fasting glucose and insulin levels. Glucose levels were higher in HF/C M than in F at MID, which vanished at END (Table 1). In contrast, insulin levels were higher in M than in F irrespective of the maternal diet at MID and remained significantly higher at END in HF/C (Table 1). In alignment, we determined lower liver insulin sensitivity (IS) (HOMA index; $p < 0.001$) and lower whole body IS (Matsuda index; $p < 0.01$), as well as impaired marker of β-cell function (AUCins:AUCglc; $p < 0.01$) in HF/C M compared to F (Table 1). These results were confirmed by oral glucose tolerance (OGTT) and insulin tolerance tests. OGTT revealed that HF/C M had higher insulin levels for similar glucose concentrations than F at both MID (Fig. 1g, h) and END (Fig. 1j, k). The ability of injected insulin to clear glucose from the circulation was comparable in all groups at MID (Fig. 1i) but reduced in HF/C M at END (Fig. 1l), indicative for peripheral IR at END (later in life), which was further supported by changes in the expression level of genes involved in insulin signaling (Supplementary Fig. S1c). Overall, our findings point out that MO impairs metabolism in M offspring later in life due to transcriptional and post-transcriptional modifications.

Metabolic complications are usually accompanied by changes in the circulating level of triglycerides (TG) and cholesterol contained into lipoproteins. While the total TG levels were independent of the maternal diet, they were significantly higher in HF/C M than in HF/C F (Fig. 1m) due to higher level of low-density lipoprotein (LDL) and high-density lipoprotein (HDL) (Fig. 1n, o) fractions. Remarkably, HDL-TG decreased in HF/C F compared to C/C F (Fig. 1o). The total cholesterol level was lower in F than in M and remained unaffected by MO in F but increased in HF/C M compared to C/C (Fig. 1p). The lipoprotein-containing cholesterol profile revealed that the LDL-Chol fraction was increased in HF/C M compared to C/C (Fig. 1q) and HDL-Chol fraction was significantly higher in M than in F in HF/C (Fig. 1r).

### MO remodels adipose tissue triglycerides composition in offspring.
Sex-dependent lipid composition in the fat depots can trigger sex differences in body fat distribution and different lipid species contribute to the development of metabolic dysfunctions in obesity[19]. Therefore, we performed in vivo proton magnetic

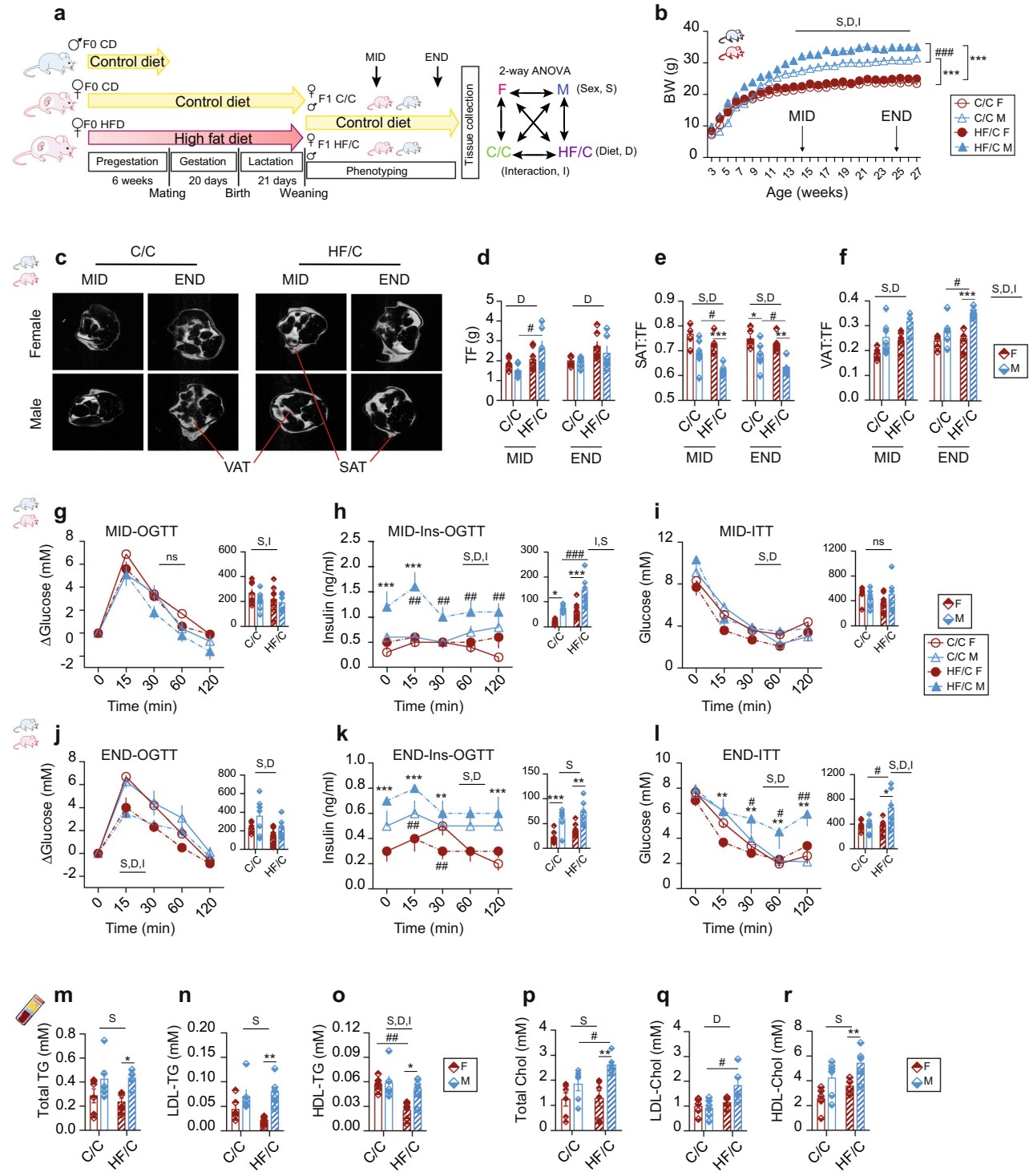

resonance spectroscopy ($^1$H-MRS) in VAT and SAT of offspring at MID and END to evaluate TG profile (Supplementary Fig. S1d). Mean chain length (MCL) of TG in VAT was similar between sexes at both time points in C/C group. In HF/C groups, although no differences were observed at MID, lower MCL was found in F than in M at END (Supplementary Fig. S1e). In contrast, in SAT, MCL was higher in F than in M in C/C and HF/C groups at MID but similar at END (Supplementary Fig. S1f). In VAT, the fraction of saturated lipids (fSL) was higher in F than in M in C/C but not in HF/C group, with no differences observed at END. The fraction of monounsaturated lipid (fMUL) was similar at MID but higher in HF/C F than in C/C F at END, whereas the

fraction of polyunsaturated lipid (fPUL) was higher in HF/C M than in C/C M at MID but similar at END (Supplementary Fig. S1g). In SAT, fSL was similar between all groups; fMUL and fPUL was highly sex and maternal diet dependent (Supplementary Fig. S1h). In sum, our in vivo data reveal a strong link between lipid composition and body fat distribution evoked by a sex-dependent adaptation to MO.

**MO-induced changes in offspring hepatic triglycerides.** In addition to diet-induced changes in the adipose tissue other organs and cell types are frequently affected. We performed

**Fig. 1 Physiological and biological adaptations to maternal obesity in F1 offspring is sex-dependent. a** Schematic overview of the experimental set-up. Dam-F0 were fed either the control diet (CD, yellow arrow) or the high-fat diet (HFD, red arrow) for 6 weeks before mating and continued to receive the same diet during gestation and lactation; male-F0 remained on CD until mating. All female (F, red bars) and male (M, blue bars) offspring remained on CD after weaning. Offspring born from CD mother (C/C, open bars) and HFD mother (HF/C, stripped bars) were monitored at 3 months of age (MID) and 6 months of age (END). Two-way ANOVA statistical comparisons are shown to the right. **b** Time course body weight (BW) curve in F (red circle; open circle for C/C and full circle in HF/C) and M (blue triangle; open triangle for C/C and full triangle in HF/C) offspring over 6 months after weaning. **c** MRI images of the lower abdominal region of F and M offspring indicating visceral (VAT) and subcutaneous (SAT) adipose tissue depots. MRI-based quantification of **d** total fat (TF), **e** SAT on TF ratio (SAT:TF), and **f** VAT on TF ratio (VAT:TF). Time course and area under the curve (AUC) of circulating glucose levels at **g** MID and **j** END and insulin levels at **h** MID and **k** END, after the glucose load given by gavage. Time course and AUC of circulating glucose levels after insulin injection at **i** MID and **l** END. Plasma total **m** triglycerides (TG) and **p** cholesterol (Chol); and AUC of **n** LDL-TG, **o** HDL-TG, **q** LDL-Chol, and **r** HDL-Chol lipoprotein fractions obtained by FPLC. For **b** C/C F ($n = 11$) and C/C M ($n = 12$), for HF/C F ($n = 11$) and HF/C M ($n = 10$). For **d–f** C/C F ($n = 6$) and C/C M ($n = 7$), for HF/C F ($n = 7$) and HF/C M ($n = 6$). For **g–l** C/C F ($n = 9$) and C/C M ($n = 8$), for HF/C F ($n = 10$) and HF/C M ($n = 7$). For **m–r** $n = 7$ per group. Data are presented as mean ± sem. Two-way ANOVA (sex (S), mother diet (D), interaction (I) between sex and diet, and (ns) for not significant) followed by Tukey's multiple comparisons test when significant ($p < 0.05$). Differences between two groups (sexes, F versus M; maternal diet C/C versus HF/C) were determined by t-test corrected for multiple comparisons using the Holm–Sidak method, with alpha = 5.000%. *M versus F and #HF/C versus C/C, $p < 0.05$; ** or ##$p < 0.01$; *** or ###$p < 0.001$.

---

**Table 1 Body weight, fasted glucose, and insulin levels and markers of insulin sensitivity and β-cell function in female (F) and male (M) offspring.**

| Diet | MIDTERM | | | | ENDTERM | | | |
| | C/C | | HF/C | | C/C | | HF/C | |
| Sex | F | M | F | M | F | M | F | M |
|---|---|---|---|---|---|---|---|---|
| Body weight (g) | 22.3 ± 0.5 | 29.2 ± 0.8 | 23.8 ± 0.5 | 32.5 ± 1.0*** | 24.2 ± 0.4 | 30.4 ± 1.0 | 25.0 ± 0.5 | 34.4 ± 0.8*** |
| Fasted glucose (mM) | 7.2 ± 0.5 | 8.1 ± 0.5 | 6.3 ± 0.3 | 9.7 ± 0.3*** | 7.6 ± 0.3 | 8.4 ± 0.6 | 6.5 ± 0.4 | 8.3 ± 0.3 |
| Fasted Insulin (ng/ml) | 0.32 ± 0.06 | 0.60 ± 0.06* | 0.55 ± 0.15 | 1.04 ± 0.17*## | 0.33 ± 0.06 | 0.45 ± 0.09 | 0.25 ± 0.04 | 0.67 ± 0.08** |
| HOMA-index | 0.10 ± 0.02 | 0.22 ± 0.02 | 0.16 ± 0.04 | 0.46 ± 0.08***## | 0.12 ± 0.02 | 0.18 ± 0.04 | 0.07 ± 0.01 | 0.25 ± 0.04** |
| Matsuda index | 4150 ± 458 | 1997 ± 185** | 3539 ± 594 | 1080 ± 210** | 4540 ± 943 | 2617 ± 431 | 6067 ± 864 | 1912 ± 308** |
| AUCins:AUCglc | 0.039 ± 0.005 | 0.062 ± 0.005** | 0.062 ± 0.011# | 0.107 ± 0.013***## | 0.034 ± 0.006 | 0.049 ± 0.005 | 0.039 ± 0.005 | 0.065 ± 0.007** |

Plasma levels and markers of insulin sensitivity in F and M offspring born from CD (C/C) or HFD (HF/C) mothers at MID and END. For glucose and insulin, animals were fasted for 6 h prior the blood sampling from the tail. For body weight, C/C F ($n = 11$) and C/C M ($n = 12$), for HF/C F ($n = 11$) and HF/C M ($n = 10$); for the rest C/C F ($n = 9$) and C/C M ($n = 8$), for HF/C F ($n = 10$) and HF/C M ($n = 7$). Data are presented as mean ± sem. Unpaired two-tailed Student's t-test was considered significant when $p < 0.05$. *M versus F and #HF/C versus C/C. * or #$p < 0.05$; ** or ##$p < 0.01$; ***$p < 0.001$. F female, M male, HOMA, homeostatic model assessment, AUC, area under the curve.

---

in vivo [1]H-MRS in offspring liver at MID and END (Fig. 2a). At MID, the fraction of total lipid mass (fLM), fSL, and fMUL were similar in both sexes and independent of MO. However, at END, fLM increased in both sexes (Fig. 2b) with no differences in liver histology (Supplementary Fig. S2a); fSL was higher in C/C M than in F and MO increased fSL in F but decreased it in M (Fig. 2c) and fMUL was significantly lower in M than in F in both diet groups (Fig. 2d). In contrast, at MID and END, fPUL decreased in HF/C F compared to C/C F and fPUL increased in HF/C M compared to C/C M at END (Fig. 2e). In conclusion, MO has major sex-dependent effects on the hepatic TG composition in offspring and indicates a solid interaction between sex and diet (two-way ANOVA) especially at END (later in life).

**LC-MS confirmed that MO induced changes in triglycerides profile in offspring liver.** The liver is the organ that predominately synthesizes and transports lipids particularly under CD. Impairment in lipid homeostasis often results in the development of liver steatosis. To further investigate the impact of MO on hepatic lipid composition in offspring, we used LC-MS to profile TG molecular species. We identified 11 different TG classes that we classified into TG low, moderate, and high, based on the total number of carbon atoms detectable in the fatty acyl (FA) chain (Fig. 2f, h). TG49, TG51, and TG53 were enriched in M compared to F in C/C and these differences disappeared in HF/C due to an increase abundance in HF/C F compared to C/C F.

TG60 abundance was significantly higher in HF/C M than in F (Fig. 2f). All moderately abundant TG were similar between sexes in C/C group. MO reduced TG56- and TG58-relative content in F compared to C/C, to a significantly lower level than in M (Fig. 2g). High-abundant TG were similar between sexes in the C/C group but TG50 and TG52 abundance was increased in HF/C F compared to C/C F to a higher level than M for TG52 (Fig. 2h). These results show that in addition to sex differences, MO selectively modulates TG classes.

Based on whether double bonds between the hydrogen atoms can be formed, FA are divided into saturated (no double bonds) and unsaturated (with double bonds). Several metabolic diseases are associated with the degree of hydrogen atom bound saturation within the FA of the TG. We, therefore, inspected the TG saturation status. In C/C, TG containing 1-double bond were higher and TG containing 3- and 4-double bonds were lower in M than in F (Fig. 2i). Surprisingly, MO severely redistributed TG saturation profile in a sex-dependent manner (Fig. 2i). In HF/C, M had less TG with 2- and 3-double bonds but four times more of TG containing 4+-double bonds than F. This was due to a complete remodeling of TG classes in HF/C F compared to C/C, while TG profile remained unaffected by the MO in M. Among the 50 TG species detected by LC-MS, we observed differences in the relative levels of 26% (13 out of 50) TG species between the sexes in C/C and 70% (35 out of 50) in HF/C (Fig. 2j and Supplementary Fig. S2b–d). Our analysis revealed that HF/C M

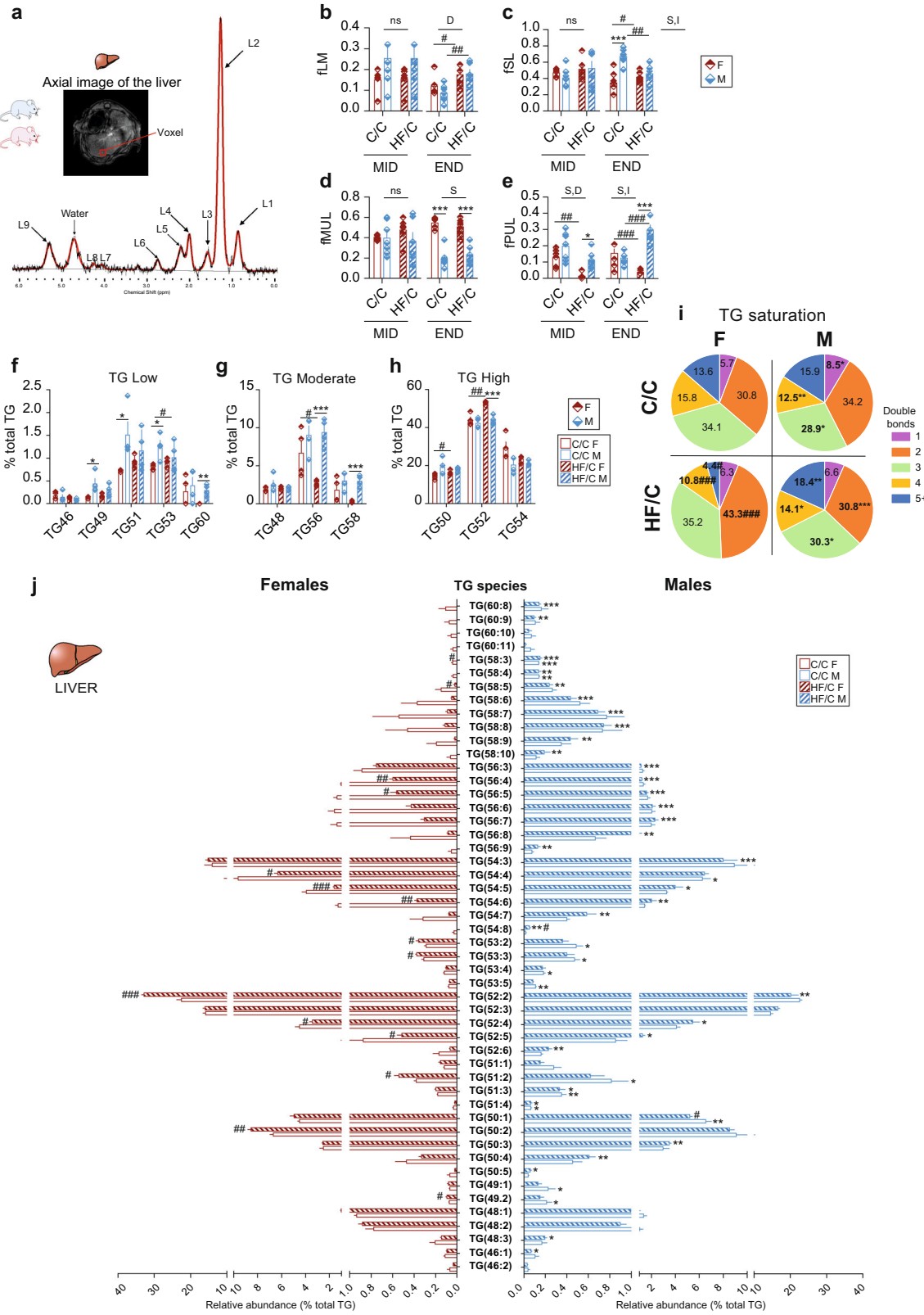

liver contained more of the very-long chain TG species when compared to F. In contrast, HF/C F exhibited higher relative content of the shorter TG species. In conclusion, MO remodels hepatic TG species and their saturation status in F but not in M.

**MO alters fatty acid composition in female but not in male offspring liver.** As bioactive signaling molecules, FA species

modulate cell activity and metabolism. We therefore explored whether the hepatic FA composition of TG and PL was sex-specific and hereby contributed to the sexual dimorphism in the metabolic adaptation of offspring born from obese mothers. Gas chromatography–mass spectrometry (GC-MS) analysis revealed no sex differences in the hepatic FA profile in C/C group. The relative amount of monounsaturated FA (MUFA, C18:1 and

**Fig. 2 Maternal obesity causes sex-specific triglyceride remodeling in liver.** Offspring born from CD mother (C/C, open bars) and HFD mother (HF/C, stripped bars) were monitored at 3 months of age (MID) and 6 months of age (END). **a** Axial image of the liver with single voxel spectroscopy and one representative proton spectrum used for in vivo quantification of the fractions of the **b** lipid mass (fLM), **c** saturated lipid (fSL), **d** monounsaturated lipid (fMUL), and **e** polyunsaturated lipid (fPUL). Relative level of TG classes evaluated by LC-MS in liver extract from F (red bars) and M (blue bars) offspring groups classified as **f** low-, **g** moderate-, and **h** high-abundant TG classes; **i** Pie charts of the TG saturation profile in C/C and HF/C F and M offspring. **j** Plot of the 50 TG species detected by LC-MS in liver extract from F (red bars) and M (blue bars) offspring in C/C (open bars) and HF/C (stripped bars). For **b–e** C/C F ($n = 6$) and C/C M ($n = 7$), for HF/C F ($n = 7$) and HF/C M ($n = 6$). For **f–j** $n = 4$ per group. Data are presented as mean ± sem. Two-way ANOVA (sex (S), mother diet (D), interaction (I) between sex and diet, and (ns) for not significant) followed by Tukey's multiple comparisons test when significant ($p < 0.05$). Differences between two groups (sexes, F versus M; maternal diet C/C versus HF/C) were determined by t-test corrected for multiple comparisons using the Holm–Sidak method, with alpha = 5.000%. *M versus F and #HF/C versus C/C, $p < 0.05$, ** or ##$p < 0.01$; *** or ###$p < 0.001$.

C16:1 species) was increased in HF/C F compared to C/C F. HF/C F liver contained more of the MUFA C18:1-ω9 and C18:1-ω11 than HF/C M, and less of saturated FA (SFA, C16:0 and C18:0), of polyunsaturated FA (PUFA, C18:2) and of very long chain (VLC)FA (C20:4 and C22:6) (Fig. 3a). Accordingly, HF/C M had relatively more of SFA and less of the 1-, 2-, and 4+-double bonds FA than F (Fig. 3b). Of note, HF/C F contained higher relative levels of C15:0, C16:1-ω9, and C18:3-ω3 FA than HF/C M, which inhibit liver steatosis and inflammation[20]. The relative level of total ω6-PUFA was higher and that of ω9-MUFA was lower in HF/C M than F (Fig. 3c), indicative of metabolic disorders including liver steatosis, higher glucose, and insulin levels[21]. MO reduced ratio of ω6:ω3 FA (Fig. 3d) and induced Δ−9 FA desaturase activity (Fig. 3e) in F, which are associated with reduced risk of many chronic diseases[22,23].

To evaluate whether the hepatic changes in total FA and TG could be explained molecularly, we investigated for transcriptional alterations in gene expression of key regulators of the lipid pathways by qPCR. Overall, the gene expression levels of the tested key regulators varied between sexes (Fig. 3f) and in response to MO (Fig. 3g). In particular, we observed 100-times higher expression level of elongase (Elovl3 promoting the formation of VLCFA), in M than in F in both dietary groups. In contrast, genes involved in stimulating desaturation of FA (Scd1, Scd2 and Fads1, Fads2) were expressed at lower levels in HF/C M than F (Fig. 3f and Supplementary Fig. S3a), which was in accordance with higher levels of VLCTG and SFA, as well as lower levels of MUFA in HF/C M than in F. Interestingly, M and F showed opposite response to MO in the expression levels of key genes of the FA synthesis and inflammatory pathways (Fig. 3g and Supplementary Fig. S3b–e). C/C F had higher expression levels of genes involved in cholesterol metabolism and lipolysis/oxidative pathways compared to M (Supplementary Fig. S3c, d). In contrast, M tended to increase the hepatic expression of genes of the inflammatory pathway in HF/C group compared to C/C. In summary, FA and TG profiles are sex-specifically regulated through altered hepatic lipid and inflammatory transcriptional regulation. This would support the notion that the in utero environment affects the transcriptional activity of metabolic genes in a sex-dependent manner.

**MO provokes profound sex-dependent changes of phospholipid species in offspring liver.** Phospholipids (PL) are critical components of cell membrane and regulate membrane integrity and fluidity. Furthermore, PL can act as signaling molecules and therefore PL concentrations must be tightly regulated. We identified 11 different PL classes in offspring liver by LC-MS including the most abundant phosphatidylcholine (PC) and phosphatidylethanolamine (PE) as well as the lysoPC (LPC) and lysoPE (LPE). A principal component analysis separated the PL classes into four distinct groups, which clustered HF/C and C/C as well as F and M (Fig. 4a).

A total of 30 PC species were identified (Supplementary Table S2 and Supplementary Fig. S4a–c). Their abundance varied between sexes by 23% (7/30) in C/C and 67% (20/30) in HF/C, respectively, and MO caused minor significant differences in PC species in M (4/30) and in F (8/30) (Fig. 4b). In C/C, F had lower proportion of PC containing 3-double bonds than M and MO remodeled PC saturation profile in F. Consequently, F showed higher proportion of 1-double bond containing PC but lower of the 2-double bonds containing PC than M in HF/C group (Fig. 4c). Among the 25 PE species detected (Supplementary Table S2 and Supplementary Fig. S4d–f), the abundance differed between sexes by 8% (2/25) in C/C and 44% (11/25) in HF/C group and MO did not significantly affect PE composition in both sexes (Fig. 4d). The PE saturation profile was similar between sexes except for the 3-double bonds containing PE that were twice higher in M than in F in HF/C group (Fig. 4e). The relative level of the total LPC class was similar between sexes in C/C group and increased with MO in both sexes althought only significantly in F (Fig. 4f). In C/C, 36% (4/11) LPC species were different between sexes and these differences disappeared in HF/C group (Fig. 4g). MO significantly induced all LPC species abundance in F except for LPC16:0 and tended to induce them in M (Fig. 4g and Supplementary Fig. S5a, b). LPC saturation profile was highly dependent of MO in F, with a reduction of the saturated LPC and an induction of the 1- and 4+-double bonds containing LPC compared to C/C F (Fig. 4h). Consequently, HF/C F liver contained more of the 1-double bond LPC ($p = 0.05$) and fewer of the 2- and 3-double bonds containing LPC than M (Fig. 4h). LPE abundancy was similar between sexes in C/C and increased in HF/C groups, significantly in F only (Fig. 4i). Relative level of LPE species increased in HF/C compared to C/C group with 75% (9/12) in F and 25% (3/12) in M (Fig. 4j and Supplementary Fig. S5c, d). No sex differences were observed in the saturation profile in the C/C group. MO reduced the saturated LPE and induced the 4+-double bonds containing LPE in F, while the 2-double bonds containing LPE in M were higher than in F (Fig. 4k). In conclusion, relative levels of PC and PE species is sex-dependent especially when born from obese mothers. In contrast, LPC and LPE abundance is similar between sexes but highly dependent on maternal diet. This indicates that the abundance and composition of the major PL classes controlling membrane integrity and trafficking in liver are sex-dependent and highly affected by MO.

**Sex-specific plasmalogen, phosphoglyceride, cardiolipin, and ceramide species partly contribute to the sexual dimorphism in the offspring metabolic adaptation to MO.** Alkylacyl PL, including plasmenyl (also called plasmalogens) and plasmanyl PC (P-PC) and plasmanyl PE (P-PE), facilitate membrane fusion. Plasmalogens can serve as endogenous antioxidants, protecting cells against reactive oxygen species (ROS). Liver synthesises plasmalogens that are further transported to other tissues by

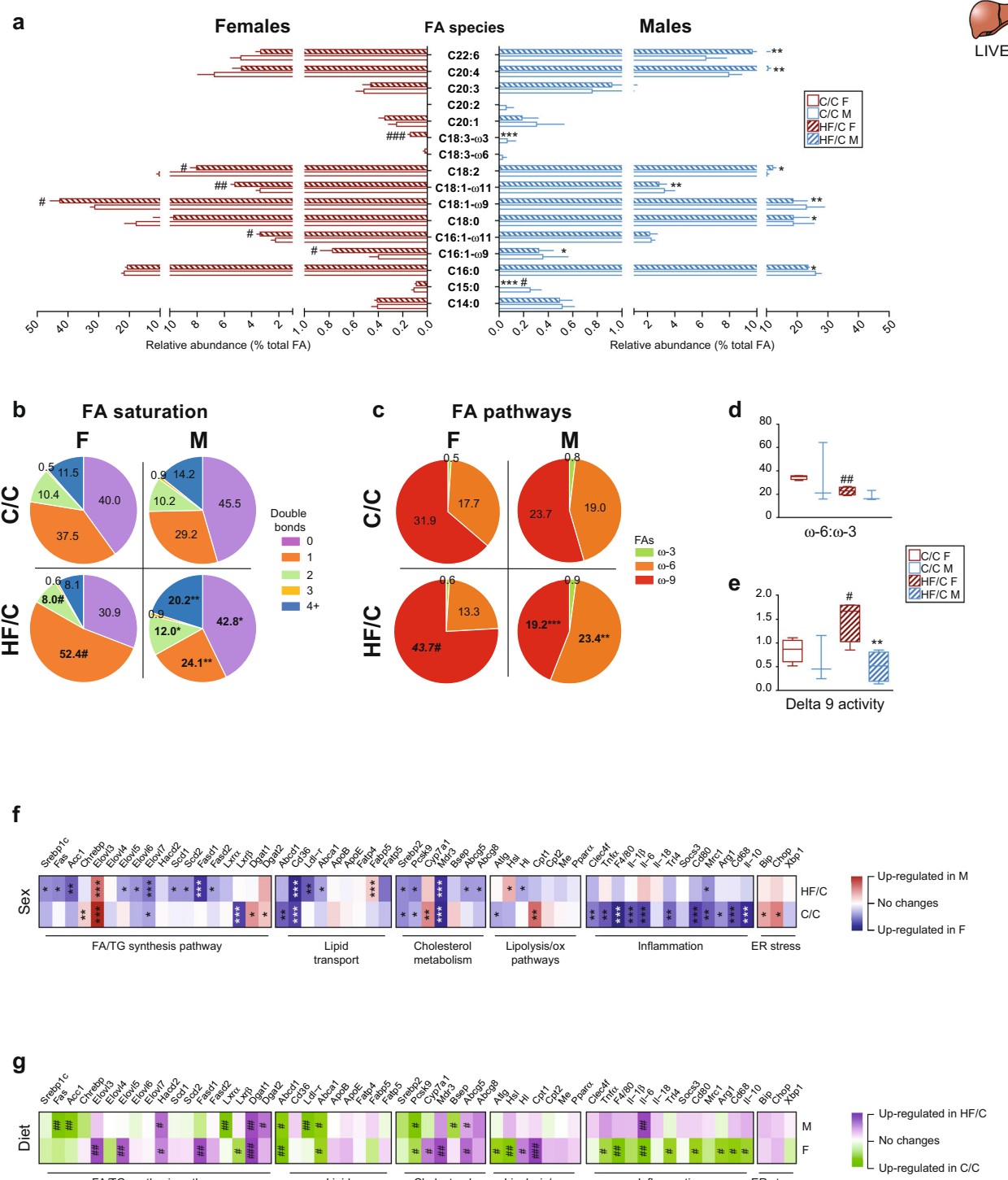

**Fig. 3 Maternal obesity alters fatty acid composition in F livers but not in M. a** Plot of the FA species contained into the TG and phospholipids detected by GC-MS in liver extract from F (red bars) and M (blue bars) offspring in C/C (open bars) and HF/C (stripped bars) groups; pie charts of the **b** FA saturation profile; **c** ω-3, ω-6 and ω-9 FA synthesis pathways; **d** ratio of ω-6 to ω-3 FA pathway; and **e** delta 9 activity of the FA synthesis pathway in C/C and HF/C F and M offspring. Heatmap showing the *p*-value of the gene expression in comparison to the group of reference (**f** for sex comparison and **g** for mother diet comparison) of the FA and TG synthesis, the lipid transport, the cholesterol metabolism, the lipolysis and lipid oxidation, the inflammatory, and the ER stress pathways. For sex comparison: M versus F in C/C, M versus F in HF/C, blue: up-regulated in females, red: up-regulated in males, white: similar between sexes. For maternal diet comparison: C/C versus HF/C within F and C/C versus HF/C within M, green: up-regulated in C/C and purple: up-regulated in HF/C, white: similar between maternal diet, where value below 0.05 is considered significant. For **a**–**e** n = 4 per group. For **f** and **g** n = 7 per group. Data are presented as mean ± sem. Differences between two groups (sexes, F versus M; maternal diet C/C versus HF/C) were determined by t-test corrected for multiple comparisons using the Holm–Sidak method, with alpha = 5.000%. *M versus F and #HF/C versus C/C (*p* < 0.05), ** or ##*p* < 0.01 and *** or ###*p* < 0.001.

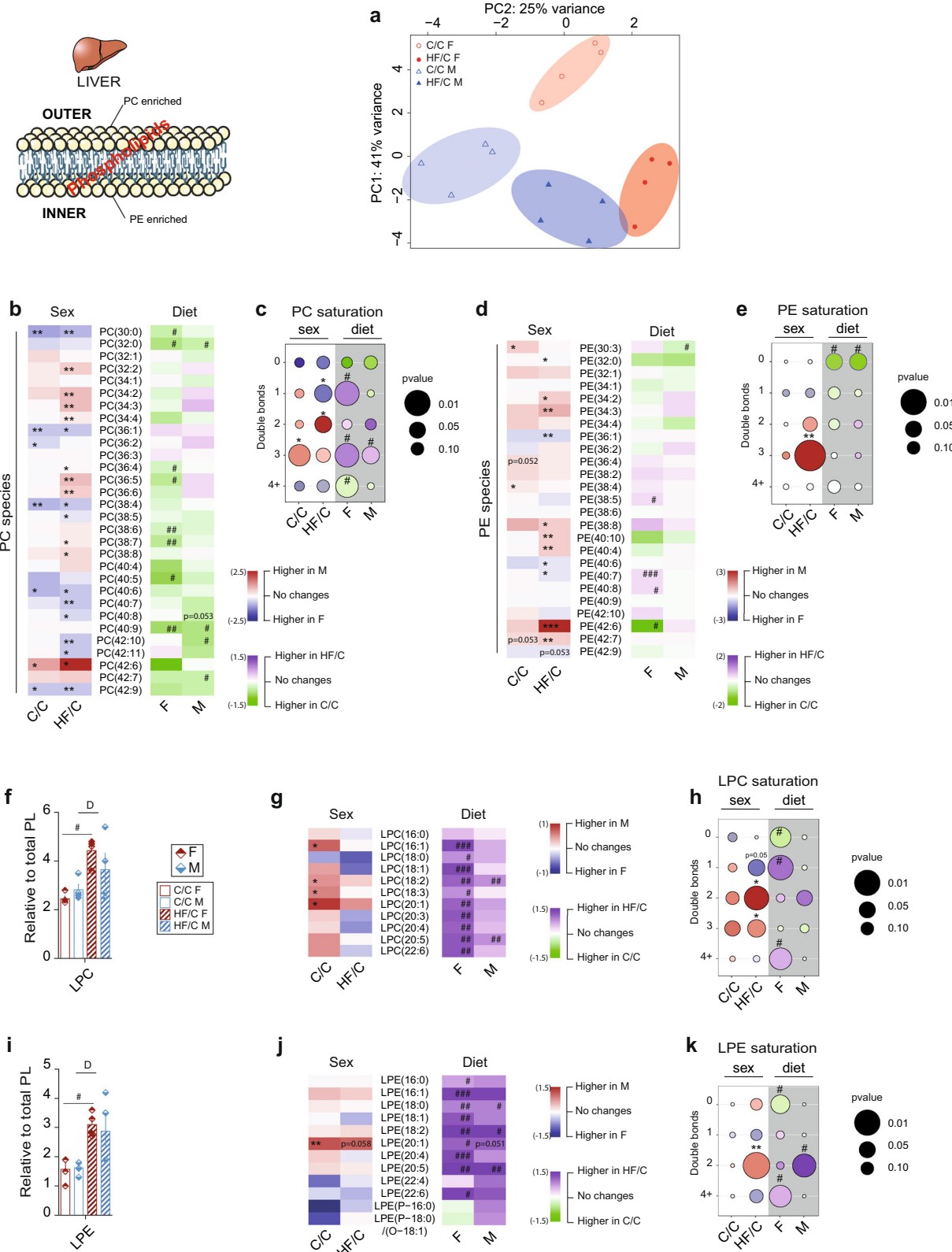

lipoproteins. We found 20% (1/5) in C/C and 40% (2/5) in HF/C P-PC species significantly different between M and F, whereas 92% (11/12) and 33% (4/12) of the P-PE species varied significantly between M and F in C/C and HF/C groups, respectively (Fig. 5a, b). Overall, relative content and percentage of P-PC and P-PE were similar between sexes in C/C but were considerably higher in HF/C F than in HF/C M (Fig. 5c, d). This difference was

largely explained by a strong remodeling of P-PE in M only. Moreover, when compared to M, F had significantly more relative amounts of the P-PE containing arachidonic acid (AA, C20:4), which functions as an antioxidant (Supplementary Table S2). In conclusion, the higher proportion of plasmalogens found in F compared to M may contribute to the metabolic protection against MO-induced IR, as compared to M.

**Fig. 4 Maternal obesity provokes profound changes of phospholipids species in liver of the offspring in a sex-dependent manner. a** PCA plot[a] of the relative hepatic phospholipid profile detected by LC-MS; heatmap of the **b** PC and **d** PE lipid species; bubble charts of the **c** PC and **e** PE saturation profile in liver extract between M and F (for sex comparison: C/C and HF/C columns, white background) and between C/C and HF/C (for maternal diet comparison: F and M columns, gray background) offspring; relative content of **f** LPC and **i** LPE lipid class; heatmap of the **g** LPC and **j** LPE lipid species; bubble charts of the **h** LPC and **k** LPE saturation profile between M and F (for sex comparison: C/C and HF/C columns, white background) and between C/C and HF/C (for maternal diet comparison: F and M columns, gray background) offspring. The size of the bubbles indicates the $p$-value. Larger bubbles correspond to higher significance where $p$-value below 0.05 is considered significant. The color of the bubble indicates the $\log_{10}$ fold change between the two group's comparison; the heatmaps were colored according to $\log_2$ fold change in comparison to the group of reference (for sex comparison: M versus F in C/C, M versus F in HF/C, blue: higher in females, red: higher in males, white: similar between sexes. For maternal diet comparison: C/C versus HF/C within F and C/C versus HF/C within M, green: higher in C/C and purple: higher in HF/C, white: no change). For **a-k** $n = 4$ per group. Data are presented as mean ± sem. Two-way ANOVA (sex (S), mother diet (D), interaction (I) between sex and diet, and (ns) for not significant) followed by Tukey's multiple comparisons test when significant ($p < 0.05$). *M versus F and #HF/C versus C/C ($p < 0.05$), ** or ##$p < 0.01$ and *** or ###$p < 0.001$. [a]The PCA plot was tilted 90° for representative purpose.

We further investigated other PL classes detected by LC-MS, including phosphatidylglycerol (PG), cardiolipin (CL), and the sphingolipid ceramide (Cer) (Supplementary Tables S2 and S3). PG abundance was similar between sexes in C/C but higher in HF/C F than M and PG species were markedly sex-dependent in HF/C (Fig. 5e, f and Supplementary Fig. S5e, f). The PG saturation profile differed between sexes in HF/C and MO resulted in a lower proportion of 1-double bond containing PG and a higher proportion of 4+-double bonds containing PG in F than in M (Fig. 5g). CL class was higher in C/C M than in F but MO increased significantly CL in F (Fig. 5h). C/C M had higher level than F of most of the CL species and HF/C F had significantly more CL compared to C/C F while the abundance remained unchanged in M (Fig. 5i and Supplementary Fig. 5g, h). No sex differences were observed in the saturation profile of CL. However, MO resulted in a shift from more 5-double bonds containing CL to less 7- and 8-double bonds containing CL in F, whereas in M, MO increased the proportion of 9+-double bonds containing CL (Fig. 5j).

The relative abundance of the Cer class was increased by 78% in HF/C M while unchanged in F compared to C/C (Fig. 5k, l). In both diet groups, F had more short chain (d34:1 and d36:1) and less long chain (d38, d40, d40, d42) Cer species, as well as less glyceroCer than M (Fig. 5l and Supplementary Fig. S6a, b). Cer containing 1-double bond increased and those with 2-double bonds decreased in HF/C F compared to C/C while no changes were observed in M (Fig. 5m). The sphingomyelin (SM) lipid class is derived from Cer, and its levels have been demonstrated as positively correlated with liver dysfunction[24]. MO generally decreased the relative abundance of SM in both sexes. Nevertheless, 50% (4/8) and 75% (6/8) of the SM species were differently abundant between sexes in C/C and HF/C, respectively (Fig. 6a, b and Supplementary Fig. S6c, d), with higher level of 3-double bonds SM in HF/C F than M (Fig. 6c). Lastly, the relative phosphatidylinositol (PI) content was higher in M than in F in both diet conditions. MO reduced the overall PI abundance in both sexes when compared to C/C (Fig. 6d). The relative content of PI species was highly sex-dependent with 77% (10/13) and 85% (11/13) of PI species being different in C/C and HF/C group, respectively (Fig. 6e and Supplementary Fig. S6e, f). MO provoked a severe reduction of PI species in F (69%, 9/13), which remained unchanged in M (Fig. 6e). C/C F had less of the PI species containing 2-, 3- and 4+-double bonds than M (Fig. 6f). In HF/C, M liver contained more PI containing 1- and 2-double bonds than F, attributable to a complete remodeling of the saturation profile of the PI in F only (Fig. 6f). Interestingly, MO blunted the lysophosphatidylinositol (LPI) relative abundance in both sexes (Fig. 6g). C/C F had more of LPI containing AA (C20:4) than M, and MO led to a remarkable reduction of LPI (C18:0) and LPI (C20:4) levels in both sexes (Fig. 6h).

In conclusion, LC-MS revealed that PL classes are highly dependent on sex and maternal diet. The profound changes affected lipid classes involved in the regulation of inflammation and liver steatosis (Cer), mitochondria function (CL) and IR (plasmalogens, SM and PI). Altering abundance of these lipid classes could be key in explaining why the F offspring is much better protected from the adverse effects of MO than M.

**MO-induced modulation of phospholipid abundance is manifested at the transcriptional level.** To link the observed PL profile with hepatic transcriptional activity, we investigated the expression level of key genes controlling PL synthesis pathways. Genes of the Kennedy pathway (PC and PE) were similarly expressed between sexes except for choline kinase (Ck) that was higher expressed in C/C M than F, but this difference disappeared in HF/C group (Fig. 6i and Supplementary Fig. S3g). However, genes involved in the phosphatidic acid and PG pathways (Agpat, Dgkδ, Gpd1, Pgs1, Ppap2α, and Gpam) and the key regulator of CL synthesis in mitochondria (Crls1) were higher expressed in F than in M in both diet groups. In contrast, the expression of genes involved in Cer synthesis (Sgm2) pathways was enhanced in M compared to F, irrespective of the maternal diet (Fig. 6i, j and Supplementary Fig. S3h). These results show that hepatic gene expression differs between sexes and their abundance is altered upon MO in a sex-dependent manner.

Overall, the results of this study established that MO modulates lipid metabolic pathways differently in F and M offspring. We demonstrate that in response to MO, PG and CL synthesis are promoted in F, which may facilitate mitochondrial function and reduce ROS. In addition, VLCFA and VLCTG syntheses are reduced, which would promote IS. Inversely, MO in males induces Cer and promotes SFA and VLCFA syntheses, which may contribute to IR, liver dysfunction (endoplasmic reticulum (ER) stress) and inflammation (Fig. 6k). Furthermore, we revealed that the metabolic adaptation to MO is sex-dependent partly due to differences in the expression level of key genes regulating fatty acid, TG and PL pathways (Fig. 6l). This pattern cannot be entirely reverted by healthy post-weaning diet.

## Discussion

As central guardian of lipid homeostasis, liver orchestrates the de novo synthesis of FA, TG, and PL and regulates their export and successive redistribution to other tissues. These processes are closely regulated by complex interactions between transcriptional factors, hormones and nuclear receptors (NR). This study reveals that pre-gestational, gestational and lactational MO influence hepatic lipid synthesis pathways differently in the F and M offspring, which is leading to more severe metabolic dysfunctions in M than in F (Fig. 6k, l). Our in vivo results demonstrate that MO

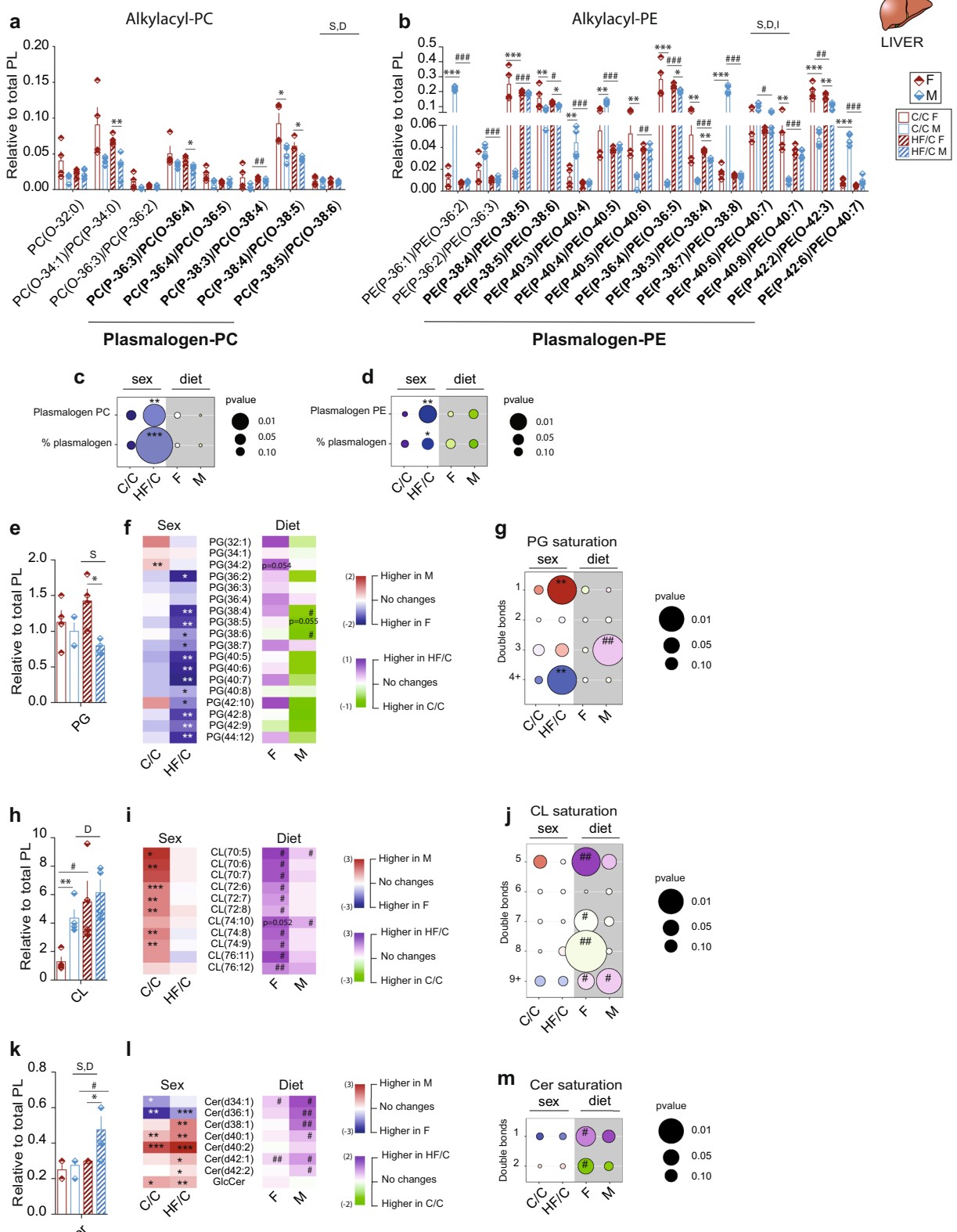

induces hepatic TG remodeling in the offspring in a sex-dependent manner, whereby fPUL levels were induced in HF/C M and reduced in HF/C F. In contrast, fSL levels were reduced in HF/C M and increased in HF/C F when compared to C/C group. When born from obese mothers, F livers contained more odd and short chain TG and MUFA, while M liver displayed more of the long chain TG and PUFA. These changes in lipid metabolism

induced by MO were irreversible and could not be rescued through post-weaning feeding of a CD. This indicates that these changes are imposed on the offspring in utero. Induction of hepatic SFA and reduction of PUFA were found in insulin-sensitive as opposed to insulin-resistant primates[25]. Increased levels of odd-chain-numbered SFA have been associated with hepatic health, while long chain FA have been linked to impaired

**Fig. 5 Plasmalogen and plasmanyl PC and PE, phosphatidylglycerol, cardiolipin, and ceramide species relative expression level in offspring.**
Relative hepatic content of **a** plasmalogen (in bold) and plasmanyl phosphatidylcholine (P-PC and O-PC) and **b** plasmalogen (in bold) and plasmanyl phosphatidylethanolamine (P-PE and O-PC) species from C/C and HF/C F and M offspring; bubble charts of the relative content and percentage of **c** P-PC and **d** P-PE between M and F (for sex comparison: C/C and HF/C columns, white background) and between C/C and HF/C (for mother diet comparison: F and M columns, gray background) offspring; Relative hepatic content and heatmaps of phosphatidylglycerides (PG) **e** class and **f** species, cardiolipin (CL) **h** class and **i** species and ceramides (Cer) **k** class and **l** species detected by LC-MS from C/C and HF/C F and M offspring; bubble charts of the **g** PG, **j** CL and **m** Cer saturation profile between M and F (for sex comparison: C/C and HF/C columns, white background) and between C/C and HF/C (for mother diet comparison: F and M columns, gray background) offspring. The size of the bubbles indicates the $p$-value. Larger bubbles correspond to higher significance where $p$-value below 0.05 is considered significant. The color of the bubbles indicates the $\log_{10}$ fold change between the two group's comparison; the heatmaps were colored according to $\log_2$ fold change in comparison to the group of reference, i.e., for sex comparison: M versus F in C/C, M versus F in HF/C, blue: higher in females, red: higher in males, white: similar between sexes and for maternal diet comparison: C/C versus HF/C within F and C/C versus HF/C within M, green: higher in C/C and purple: higher in HF/C, white: no change). For **a-m** $n = 4$ per group. Data are presented as mean ± sem. Two-way ANOVA (sex (S), mother diet (D), interaction (I) between sex and diet, and (ns) for not significant) followed by Tukey's multiple comparisons test when significant ($p < 0.05$). *M versus F and #HF/C versus C/C ($p < 0.05$), ** or ##$p < 0.01$ and *** or ###$p < 0.001$.

liver function[26] due to increase of re-esterification instead of oxidation of FA[27]. These new findings would support the idea that F regulate and metabolize lipids differently than M and give F stronger protection from metabolic complications when born from obese mothers.

HF/C M weighed more than C/C M despite no measurable changes in TF and impaired IS. M displayed higher insulin levels than F but fasting insulin levels and the ratio of AUCins:AUCglc during the OGTT were reduced between MID and END in M. This demonstrate an improved liver metabolism in M between MID and END and would suggest that diet reversal post-weaning could partly rescue M from in utero changes. However, peripheral IS remained impaired in HF/C M at END, which could explain common diabetic phenotypes later in life, in line with the recent study[28].

While obesity and related diseases in most humans have multifactorial etiologies, recent discoveries in lipid droplet biology have begun to shed light on mechanisms that contribute to these metabolic diseases[29]. Cytoplasmic lipid droplets are dynamic organelles with inter-organellar communication with several cytoplasmic cellular entities including ER and mitochondria. Therefore, imbalance in lipid droplet function has been associated with various human diseases, especially in obesity. Indeed, saturation or remodeling of TG molecular species in lipid droplets leads to severe cardiovascular diseases in humans. A number of studies indicate that diet-derived signaling lipid molecules such as plasmalogens and Cer act to positively or negatively regulate IR and inflammation in several tissues, including liver and adipose tissue[12,30] which may alter other tissue functions as a result[31]. Lipid species have recently been identified as sex-specific key components in the development of metabolic dysfunctions in obesity[11,12,31]. In this study, F offspring born from obese mothers remodeled lipid species toward more CL and plasmalogen, as a protective response[30,32]. In contrast, we observed reprogramming of the hepatic lipid synthesis in HF/C M toward more Cer lipid species, which induced ER stress, altered insulin sensitivity and liver homeostasis[33,34]. It remains to be clarified how HFD before and during pregnancy until weaning reprograms the synthesis of different lipid species in M and F. It is likely that estrogens contribute to the dietary metabolic changes in women and men, which likely involves a complex interaction between organs, sex hormone receptors, and gene programming[14,15].

Because PL are major membrane components, and thus important for maintaining membrane integrity, the dysregulation of PL turnover and trafficking within the membrane should affect on cellular functions. HF/C F liver contained a lower proportion of SFA-containing PC and a higher proportion of MUFA-containing PC together with increased LPC and LPE production,

which may indicate a protective response from HFD in utero environment in F[35,36]. In addition, HF/C F livers contained more P-PC and P-PE, which are key structural components of the cell and protect against hepatic steatosis[30], male reproduction[37], and cardiovascular diseases including IR[38]. Conversely, HF/C M liver contained more Cer and PI and reduced PG species than F; possibly through a complex sex dimorphic NR crosstalk as reviewed[14].

The mechanism by which neonatal distresses translate into pathological imprinting remains poorly understood. Gene expression analysis revealed important modifications in F and M offspring born from obese mothers compared to C/C, probably due to changes in DNA methylation in promoter regions of metabolic genes during HFD-gestation[39,40] and lactation[41,42] compared to control mothers. However, gene expression profiling in M and F showed differences in adaptation to MO, which mainly affected genes involved in inflammatory, lipolysis and oxidative pathways, and PL and ER stress pathways. These differences would imply that the methylation status changes in a sex-specific manner in the in utero environment, which is in line with other studies[43]. These findings point out the need to develop and design sex-balanced experiments to better understand metabolic adaptation to nutritional disorders.

MO likely induces a panoply of events that sculpt the hepatic lipidome in a sex-dependent manner. The mechanisms by which neonatal perturbations translate into sex-specific pathological imprinting remain poorly understood. Our findings provide a rationale to explore the role of sex hormones in the regulation of lipid pathways in the MO offspring. We show that liver function rewired upon neonatal overfeeding in a sex-specific manner in offspring due to sex-dependent transcriptional and post-transcriptional modifications of the lipid network. It is interesting to note that F may have specific regulators that protect from metabolic diseases. We speculate that this regulation is dependent on sex hormones and sex hormone receptors. However, the exact molecular interactions remain to be assessed. Understanding these molecular mechanisms may provide new targets for intervention in order to prevent metabolic disease development. Furthermore, identifying sex-specific drivers of metabolic pathways in the neonatal environment is pivotal to offer appropriate preventive measures later in life.

## Methods
**Mice and diet.** All animal procedures were approved by the local Ethical Committee of the Swedish National Board of Animal Experiments. Virgin C57Bl6/J female dams and male sires were received at 4 weeks of age. F0 dams were housed in pairs in six different cages and fed either the CD (D12450H, Research Diets, NJ, USA; 10% kcal fat from soybean oil and lard; $n = 6$, F0-CD) or the HFD (D12451, Research Diets, NJ, USA; 45% kcal fat from soybean oil and lard; $n = 6$, F0-HFD)

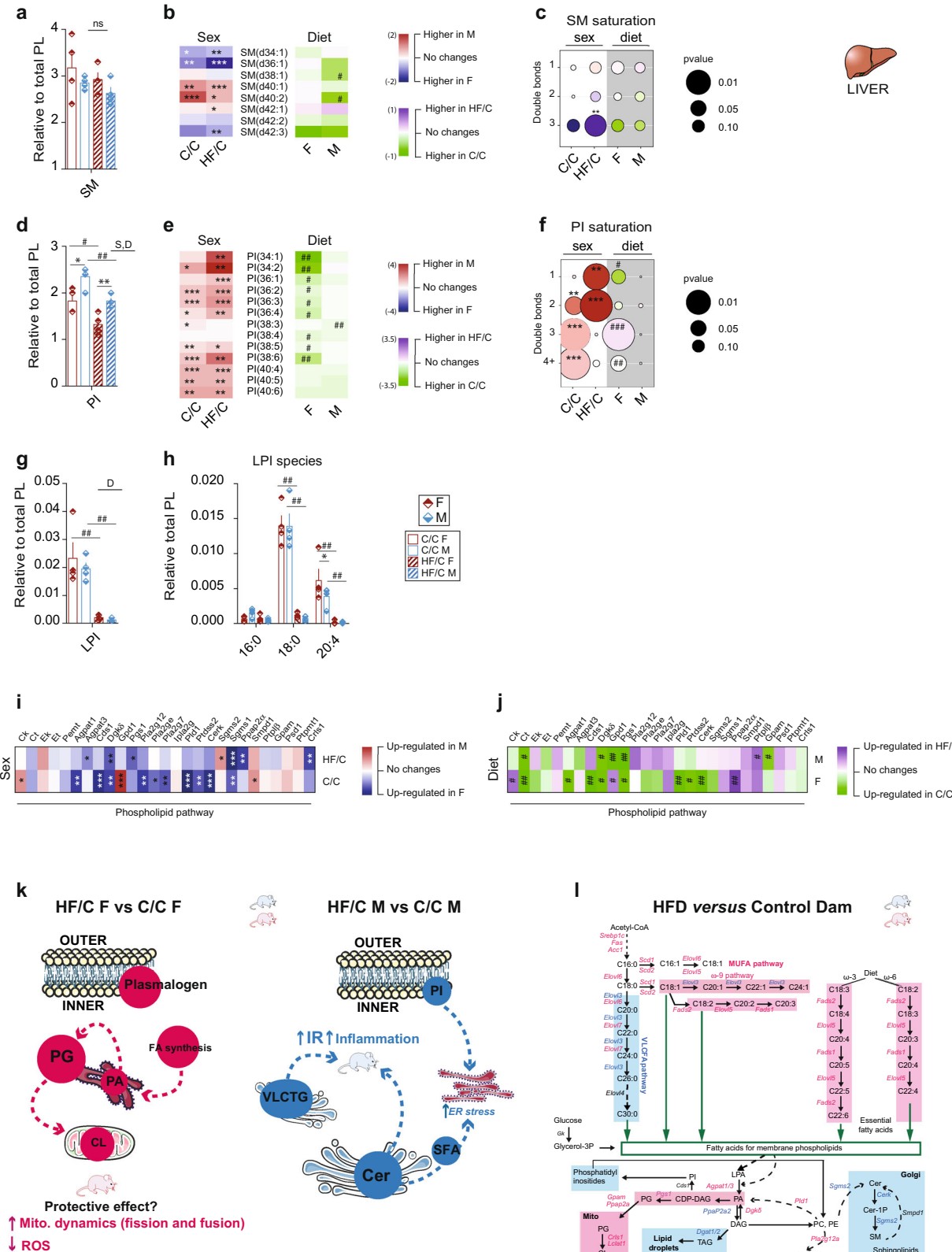

for 6 weeks before mating. Sires remained on CD until sacrifice. After 6 weeks of their respective diet two F0 dams were mated with one F0 sire. During this short mating period (up to 5 days), sires were on the same HFD as dams in the group (experimental unit). The sires spermatozoa were unlikely affected by the HFD given a general sperm maturation time of approximately 35 days[44]. After mating, F0 males and pregnant dams were separated. F0 dams were continuously exposed to their respective diets throughout pregnancy and until the end of the lactation period. The F1 offspring were weaned at postnatal day 21 (P21). Afterwards F1 males and females were sex-separated, three to five animals were housed per cage and fed with CD until the end of the study (Fig. 1a). To simplify the naming convention, the group of offspring born from HFD fed dams were named HF/C (for HFD F0 dam and CD F1 offspring) and the group of offspring born from CD fed mother named C/C (for CD mother and CD offspring). All mice were housed in a 23 °C temperature-controlled 12 h light/dark room, with free access to water and food unless specified. BW was recorded weekly throughout the study in all groups. Average food intake in offspring was recorded twice a week for 3 weeks in

**Fig. 6 Hepatic phospholipid classes are impacted by maternal diet and are sex-dependent.** Relative hepatic content of **a** sphingomyelin (SM), **d** phosphatidylinositol (PI), and **g** LysoPI (LPI) classes; Heatmaps showing the **b** SM and **e** PI species and **h** bar chart showing the LPI species; bubble chart of the **c** SM and **f** PI saturation profile in liver extract between M and F (for sex comparison: C/C and HF/C columns, white background) and between C/C and HF/C (for mother diet comparison: F and M columns, gray background) offspring; heatmaps showing the *p*-value of the gene expression in comparison to the group of reference of the phospholipid pathway between **i** sex and **j** mother diet; **k** graphical representation of the main findings of the metabolic adaptation to maternal obesity in F and M compared to CD mother; and **l** schematic overview of the fatty acid, phospholipid, and sphingolipid pathways in the liver of offspring born from HFD mother compared to CD mother. The enzymes involved in the lipid biosynthesis pathways are colored according to the expression level of the coding gene (letter color pink: higher in F, blue: higher in M, black: similar between sexes). The color of the background indicates the relative abundance of the end lipid product detected by LC-MS, pink background: higher in F, light blue background: higher in M. The size of the bubbles indicates the *p* value. Larger bubbles correspond to higher significance where *p* value below 0.05 is considered significant. The color of the bubbles indicates the log10 fold change between the two group's comparison; the heatmaps were colored according to log2 fold change in comparison to the group of reference (for sex comparison: M versus F in C/C, M versus F in HF/C, blue: higher in females, red: higher in males, white: similar between sexes. For maternal diet comparison: C/C versus HF/C within F and C/C versus HF/C within M, green: higher in C/C and purple: higher in HF/C, white: no change). For **a–h** $n = 4$ per group. For **i** and **j** $n = 7$ per group. Data are presented as mean ± sem. Two-way ANOVA (sex (S), mother diet (D), interaction (I) between sex and diet, and (ns) for not significant) followed by Tukey's multiple comparisons test when significant ($p < 0.05$). *M versus F and #HF/C versus C/C ($p < 0.05$), ** or ##$p < 0.01$ and *** or ###$p < 0.001$.

four different cages containing grouped mice ($n = 3$–5 animals per cage) around postnatal day (P110) of age and at least 1 week after recovering of in vivo experiments. We then calculated the average food intake per cage during the 3 experimental weeks. We reported it to the average food intake per mouse according to the number of animals in the cage.

**In vivo magnetic resonance imaging (MRI).** Animals were anesthetized using isoflurane (4% for sleep induction and ~2% for sleep maintenance) in a 3:7 mixture of oxygen and air, before being positioned prone in the MR-compatible animal holder. Respiration was monitored during scanning (SA-instruments, Stony Brook, NY, USA). Core body temperature was maintained at 37 °C during scanning using a warm air system (SA-instruments, Stony Brook, NY, USA).

The MRI experiments ($n = 6$–7 per group) were conducted using a 9.4 T horizontal bore magnet (Varian, Yarnton, UK) equipped with a 40-mm millipede coil, as previously detailed[45]. All images were collected on the matrix size $256 \times 96$ and a field-of-view of $51.2 \times 51.2$ mm$^2$. Fiji software (http://fiji.sc) was used to compute the volume of total fat (TF), visceral fat (VAT), and total subcutaneous fat (SAT). VAT was calculated as the difference between the TF signal and SAT signal in the abdominal region. MRI experiments were performed on the same mouse (F1) at P80 (MID) and P150 (END).

**In vivo localized proton magnetic resonance spectra (¹H-MRS).** As for the MRI scanning, animals were anesthetized using isoflurane, respiration was monitored, and core body temperature maintained at 37 °C during scanning. In addition, heart beats were recorded using an electrocardiogram system. ¹H-MRS from the liver ($n = 5$–7 per group) were acquired from a $2 \times 2 \times 2$ mm$^3$ voxel localized in the left lobe with excitation synchronized to the first R-wave within the expiration period, as detailed[17,18].

Localized ¹H-MRS from visceral and subcutaneous fat depots were acquired from $2 \times 1.5 \times 1.5$ mm$^3$ voxels positioned in the upper gonadal abdominal fat (as representative of visceral fat) and at in inguinal abdominal fat (as representative of the subcutaneous fat) (Supplementary Fig. 1e). Point Resolved Spectroscopy was used as primary pule sequence[46] with the following parameters: time to echo 15 ms, sweep width 8013 Hz, number of excitations 16, refocusing pulses 1.6 ms mao pulses with a bandwidth nominal bandwidth of 2936 Hz. The carrier frequency was shifted by 2.4 ppm, relative to water to compensate for chemical shift displacement. Outer volume suppression with a gap of 1 mm to the voxel by through slab selective hyperbolic secant pulses followed by ensuing spoiler gradients was used. The excitation was triggered to the first R-wave of the ECG within the expiration period occurring at least 3 s after the preceding excitation. All spectroscopy data were processed using the LCModel analysis software (http://s-provencher.com/pub/LCModel/manual.pdf). "Liver 9" for liver spectrum and "lipid 6" for adipose spectrum were used as a base with all signals occurring in the spectral range of 0–7 ppm (water resonance at 4.7 ppm) simulated in LCModel. All concentrations were derived from the area of the resonance peaks of the individual metabolites. Only the fitting results with an estimated standard deviation of less than 20% were further analyzed. ¹H-MRS spectra revealed nine lipid signals (peaks) in the mouse liver. Peak assignments were based on published data[47,48]. As for the MRI, ¹H-MRS experiments were repeated twice on the same animal at MID and END.

**In vivo metabolic tolerance tests.** At P80 (MID) and P150 (END), F1 mice were fasted for 6 h prior to the glucose test and 4 h prior to the insulin test ($n = 7$–10 animals per sex) and performed as detailed[49]. Matsuda index (whole body IS index) and direct measurement of hepatic IR (HOMA index) were calculated as described[50,51]. Briefly, Matsuda index was calculated as follows $= 100/(\sqrt{[G_0 \times I_0 \times G_{mean} \times I_{mean}]})$, the suffix

*mean* indicates the average value of glucose and insulin concentration measured during the whole length of the glucose test. HOMA index was calculated as HOMA $= (I_0 \times G_0)/22.5$. Evaluation of β-cell function was calculated by dividing the area under the curve (AUC) of insulin and glucose levels during the glucose test (AUCins:AUCglc).

Prior to sacrifice, mice were fasted for 2 h and anesthetized with 4% isoflurane. Blood glucose level was measured with a OneTouch Ultra glucometer (AccuChek Sensor, Roche Diagnostics). Subsequently, mice were exsanguinated via cardiac puncture and blood saved for plasma analysis. The whole liver was quickly removed and washed into PBS. A piece of left lobe of the liver was collected, fresh-frozen into liquid nitrogen and stored at −80 °C until further analysis. The rest of the liver was fixed into paraformaldehyde for histology.

**Liver histology.** For hematoxylin and eosin staining, the livers were fixed in embedding medium (O.C.T.) for frozen tissue and immediately frozen on dryice. Blocks were sectioned and stained according to standard histological procedures.

**Biochemical analysis of plasma.** Within 15 min after blood collection, plasma was separated by centrifugation (15 min at 2000 RPM) and aliquoted to avoid repeated freeze/thaw. Plasma total TG (Total-TG) and total cholesterol (Total-C) were measured by enzymatic assay using commercially available kits (Roche Diagnostics GmbH, Mannheim and mti Diagnostic GmbH, Idstein, Germany). Cholesterol lipoprotein fractions in serum were determined as described[52]. Briefly, sera (1 μl) from each individual mouse were separated by size exclusion chromatography using a Superose and PC 3.2/30 column (Pharmacia Biotech, Uppsala, Sweden). Reagent (Roche Diagnostic, Mannheim, Germany) was directly infused into the eluate online and the absorbance was measured. The concentration of the different lipoprotein fractions was calculated from the AUC of the elution profiles by using the EZChrom Elite software (Scientific Software; Agilent Technologies, Santa Clara, CA, USA).

**Lipidomic analysis of liver extract.** *Fatty acid analysis using gas chromatography–mass spectrometry (GC-MS).* Total lipid extracts were obtained using a modified Bligh and Dyer method[53] and after transmethylation, the fatty acids were analyzed by gas chromatography followed by mass spectrometry (GC-MS)[54,55]. Aliquots of the lipid extracts corresponding to 2.5 μg of total PL were transferred into glass tubes and dried under a nitrogen stream. Resulting lipid films were dissolved in 1 mL of *n*-hexane containing a C19:0 as internal standard (1.03 μg mL$^{-1}$, CAS number 1731-94-8, Merck, Darmstadt, Germany) with addition of 200 μL of a solution of potassium hydroxide (KOH, 2 M) in methanol, followed by 2 min vortex. Then, 2 ml of a saturated solution of sodium chloride (NaCl) was added, and the resulting mixture was centrifuged for 5 min at 626× *g* for phase separation. Cholesterol was removed from the organic phase according to the Lipid Web protocol (https://lipidhome.co.uk/ms/basics/msmeprep/index.htm). A 1-cm silica column in a pipette tip with wool was pre-conditioned with 5 ml of hexane (high-performance liquid chromatography (HPLC) grade). Methyl esters were added to the top of the tip and recovered by elution with hexane:diethyl ether (95:5, v/v, 3 ml), and thereafter dried under a nitrogen current. Fatty acid methyl esters were dissolved in 100 μl and 2.0 μl were injected in GC-MS (Agilent Technologies 8860 GC System, Santa Clara, CA, USA). GC-MS was equipped with a DB-FFAP column (30 m long, 0.32 mm internal diameter, and 0.25 μm film thickness (J & W Scientific, Folsom, CA, USA)). The GC equipment was connected to an Agilent 5977B Mass Selective Detector operating with an electron impact mode at 70 eV and scanning the range *m/z* 50–550 in a 1-s cycle in a full scan mode acquisition. Oven temperature was programmed from an initial temperature of 58 °C for 2 min, a linear increase to 160 °C at 25 °C min−1, followed by linear increase at 2 °C min$^{-1}$ to 210 °C, then at 20 °C min$^{-1}$ to 225 °C, standing at 225 °C for 20 min. Injector

and detector temperatures were set to 220 and 230 °C, respectively. Helium was used as the carrier gas at a flow rate of 1.4 ml min$^{-1}$. GCMS5977B/Enhanced MassHunter software was used for data acquisition. To identify fatty acids (FA), the acquired data were analyzed using the qualitative data analysis software Agilent MassHunter Qualitative Analysis 10.0. FA identification was performed by MS spectrum comparison with the chemical database NIST library and confirmed with the literature.

The Omega-3 ($\omega$−3) index was evaluated as the percent of C22:6ω3 FA content in the liver extract. The total ω−3 content was calculated as the summed total of ω−3 PUFA of C18:3ω−3, C20:5ω−3, C22:5ω−3, and C22:6ω−3. Total ω−6 content was calculated as the summed total of C18:2ω−6, C18:3ω−6, C20:2ω−6, C20:3ω−6, and C20:4ω−6 contents. Total ω9-MUFA were calculated as the summed of C16:1ω−9 and C18:1ω−9 contents.

*Reagents/Chemicals for LC-MS analysis.* PL internal standards 1,2-dimyristoyl-*sn*-glycero-3-phosphocholine (dMPC), 1-nonadecanoyl-2-hydroxy-*sn*-glycero-3-phosphocholine (LPC), 1,2-dimyristoyl-*sn*-glycero-3-phosphoethanolamine (dMPE), N-palmitoyl-D-*erythro*-sphingosylphosphorylcholine (NPSM – SM d18:1/17:0), N-heptadecanoyl-D-erythro-sphingosine (Cer d18:1/17:0), 1,2-dimyristoyl-*sn*-glycero-3-phospho-(10-rac-)glycerol (dMPG), 1,2-dimyristoyl-*sn*-glycero-3-phospho-L-serine (dMPS), tetramyristoylcardiolipin (TMCL), 1,2-dimyristoyl-*sn*-glycero-3-phosphate (dMPA), and 1,2-dipalmitoyl-*sn*-glycero-3-phosphatidylinositol (dPPI) were purchased from Avanti Polar Lipids, Inc. (Alabaster, AL, USA). HPLC grade dichloromethane, methanol, and acetonitrile were purchased from Fisher scientific (Leicestershire, UK). All the reagents and chemicals used were of the highest grade of purity commercially available and were used without further purification. The water was of Milli-Q purity (Synergy1, Millipore Corporation, Billerica, MA, USA).

## LC-MS analysis of phospholipids (PL), sphingolipids (SL), and triglycerides (TG).

Total lipid extracts from the left lobe of the liver were separated using a HPLC system (Ultimate 3000 Dionex, Thermo Fisher Scientific, Bremen, Germany) with an autosampler coupled online to a Q-Exactive hybrid quadrupole Orbitrap mass spectrometer (Thermo Fisher Scientific), adapted from refs. [53,56]. Briefly, the solvent system consisted of two mobile phases: mobile phase A (ACN/MeOH/water 50:25:25 (v/v/v) with 2.5 mM ammonium acetate) and mobile phase B (ACN/MeOH 60/40 (v/v) with 2.5 mM ammonium acetate). Initially, 10% of mobile phase A was held isocratically for 2 min, followed by a linear increase to 90% of A within 13 min and a maintenance period of 2 min, returning to the initial conditions in 3 min, followed by a re-equilibration period of 10 min prior to the next injection. Five µg of PL from total lipid extracts was mixed with 4 µL of PL standards mix (dMPC: 0.02 µg, dMPE: 0.02 µg, SM: 0.02 µg, LPC: 0.02 µg, TMCL: 0.08 µg, dPPI: 0.08 µg, dMPG: 0.012 µg, dMPS: 0.04 µg, Cer: 0.04 µg, dMPA: 0.08 µg) and 91 µL of solvent system (90% of eluent B and 10% of eluent A). Five µL of each dilution were introduced into the AscentisSi column (10 cm × 1 mm, 3 µm, Sigma-Aldrich, Darmstadt, Germany) with a flow rate of 50 µL min$^{-1}$. The temperature of the column oven was maintained at 35 °C. The mass spectrometer with Orbitrap technology operated in positive (electrospray voltage 3.0 kV) and negative (electrospray voltage −2.7 kV) ion modes with a capillary temperature of 250 °C, a sheath gas flow of 15 U, a high resolution of 70,000, and AGC target 1e6. In MS/MS experiments, cycles consisted of one full scan mass spectrum and ten data-dependent MS/MS scans (resolution of 17,500 and AGC target of 1e5), acquired in each polarity. Cycles were repeated continuously throughout the experiments with the dynamic exclusion of 60 s and an intensity threshold of 2e4. Normalized collisional energy ranged between 20, 25, and 30 eV.

Spectra were analyzed in positive and negative mode, depending on the lipid class. Cer, glucosylceramides (GlcCer), PE, LPE, PC, LPC, and SM were analyzed in the LC-MS spectra in the positive ion mode, and identified as [M+H]$^+$ ions, while CL, phosphatidylserine (PS), PI, LPI, and PG species were analyzed in negative ion mode, and identified as [M−H]$^-$ ions. Molecular species of triacylglycerol (TG) were also analyzed in positive ion mode as [M+NH$_4$]$^+$ ions. Data acquisition was carried out using the Xcalibur data system (V3.3, Thermo Fisher Scientific, USA). The mass spectra were processed and integrated through the MZmine software (v2.32)[57]. This software allows for filtering and smoothing, peak detection, alignment and integration, and assignment against an in-house database, which contains information on the exact mass and retention time for each PL, Cer, and TG molecular species. During the processing of the data by MZmine, only the peaks with raw intensity higher than 1e4 and within 5 ppm deviation from the lipid exact mass were considered. The identification of each lipid species was validated by analysis of the LC-MS/MS spectra. The product ion at *m/z* 184.07 (C$_5$H$_{15}$NO$_4$P), corresponding to phosphocholine polar head group, observed in the MS/MS spectra of the [M + H]$^+$ ions allowed to pinpoint the structural features of PC, LPC, and SM molecular species under MS/MS conditions[53], which were further differentiated based on *m/z* values of precursor ions and characteristic retention times. PE and LPE molecular species ([M+H]$^+$ ions) were identified by MS/MS based on the typical neutral loss of 141 Da (C$_2$H$_8$NO$_4$P), corresponding to phosphoethanolamine polar head group. These two classes were also differentiated based on *m/z* values of precursor ions and characteristic retention times. The [M+H]$^+$ ions of Cer and GlcCer molecular species were identified by the presence of product ions of sphingosine backbone in MS/MS spectra, such as ions at *m/z* 264.27 (C$_{18}$H$_{34}$N) and 282.28 (C$_{18}$H$_{36}$NO) for sphingosine d18:1[58], together with

the information on *m/z* values of precursor ions and characteristic retention times. The PG molecular species were identified by the [M−H]$^-$ ions and based on the product ions identified in the corresponding MS/MS spectra, namely the product ions at *m/z* 152.99 (C$_3$H$_6$O$_5$P) and 171.01 (C$_3$H$_8$O$_6$P). PI and LPI, also identified as [M−H]− ions, were confirmed the product ions at *m/z* 223.00 (C$_6$H$_8$O$_7$P), 241.01 (C$_6$H$_{10}$O$_8$P), 297.04 (C$_9$H$_{14}$O$_9$P), and 315.05 (C$_9$H$_{16}$O$_{10}$P), which all derived from phosphoinositol polar head group[53,59]. The [M−H]$^-$ ions of PS molecular species were identified based on product ions at *m/z* 152.99 (C$_3$H$_6$O$_5$P) in MS/MS spectra, retention time and *m/z* values of precursor ions. CL molecular species ([M−H]$^-$ ions) were characterized by MS/MS with identification of ions at *m/z* 152.99 (C$_3$H$_6$O$_5$P), carboxylate anions of FA chains (RCOO$^-$), product ions corresponding to phosphatidic acid anion, and phosphatidic acid anion plus 136 Da as previously reported[59]. Negative ion mode MS/MS data were used to identify the fatty acid carboxylate anions RCOO$^-$, which allowed the assignment of the FA chains esterified to the PL precursor. The MS/MS spectra of [M+NH$_4$]$^+$ ions of TGs allowed the assignment of the FA substituents on the glycerol backbone[60].

## Unsupervised clustering.
The raw data matrix of the lipid spectra was distributed column-wise by sample IDs and row-wise by PL names. The TMM method was used to normalize using between samples[61]. Unsupervised clustering was then performed using the PCA plot option in R. The PCA plot is based on the two most variant dimensions in which the PL parameters with duplicated data are filtered out.

## Hepatic RNA isolation.
Total RNA was extracted using QIAGEN miRNeasy Minit Kit (Qiagen) and DNase treated with RNase-Free DNase Set (79254) for digesting possible DNA traces. mRNA expression levels were quantified as described[18] and normalized to the control female (C/C F) group. Relative gene expression changes were calculated using *β-actin* and *Gapdh* as internal references. The primer sequences are listed in the Supplementary Table S1.

## Statistics and reproducibility.
Details about experimental design and statistics used for data analyses performed in this study were given in the respective section under results and in the figure legends. Statistical analysis was carried out using GraphPad Prism 7 software. All data are expressed as mean ± sem. Differences between offspring sex and maternal diet groups (C/C F, C/C M, HF/C F, and HF/C M) were determined using two-way ANOVA with diet (D) and sex (S) as independent variables, followed by Tukey's multiple comparison post hoc test when significant ($p < 0.05$). Differences between two groups (sexes, F versus M; maternal diet, C/C versus HF/C) were determined by unpaired two-tailed Student's *t*-test corrected for multiple comparisons using the Holm–Sidak method, with alpha = 5.000%. *$p < 0.05$ M versus F and #$p < 0.05$ HF/C versus C/C were considered significant. ** or ##$p < 0.01$; *** or ###$p < 0.001$.

## Data availability
The raw, processed and normalized lipidomic data (MetaData.txt, Phospholipids_normalized.txt, Phospholipids_raw.txt, Triglycerides_normalized.txt andTriglycerides_raw.txt) have been deposited on the public repository https://figshare.com/s/cc293caa383b439ea0be. Source data associated with Figs. 2f, j, 3a, e, 4a, k, 5a, m and Fig. 6a, h can be found in Supplementary Tables S2 and S3 and on the public repository https://figshare.com/s/cc293caa383b439ea0be. All other data included in the publication are available from the authors upon reasonable request.

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

## Acknowledgements

The MRI and MRS experiments were performed at the Department of Comparative Medicine/Karolinska Experimental Research and Imaging Centre at Karolinska University Hospital, Solna, Sweden. We thank Peter Damberg and Sahar Nikkhou Aski for excellent assistance to develop the sequence for proton-magnetic resonance spectroscopy in the liver and in the adipose tissue. We also thank Ingela Arvidsson for excellent assistance for lipoprotein plasma analysis. This work and MKA were supported by the Novo Nordisk Foundation (NNF14OC0010705), by the Lisa and Johan Grönbergs Foundation (2019-00173) and by AstraZeneca (ICMC). B.A. is supported by Swedish Research Council, Swedish Heart-Lung Foundation, Stockholm County Council/Karolinska Institutet (ALF). C.K. is supported by the Knut & Alice Wallenberg foundation (KAW 2016.0174) and Ruth & Richard Julin foundation (2018–00328). L.A.H. is supported grants from FCT—Fundação para a Ciência e a Tecnologia (UID/BIM/04501/2019, UID/BIM/04501/2020), CCDRC (CENTRO-01-0145-FEDER-000003). M.R.D. is supported by CESAM (UIDP/50017/2020+UIDB/50017/2020), QOPNA (FCT UID/QUI/00062/2019), and LAQV/REQUIMTE (UIDB/50006/2020). Fetus in Fig. 1a was created by Servier Medical Art "new born mouse." In Fig. 4, cell membrane was designed using Servier Medical Art "cell membrane" http://smart.servier.com/. Figure 6k was designed using Servier Medical Art "cell membrane," "mitochondria," "golgi apparatus," and "rough endoplansmic reticulum" http://smart.servier.com/. Open Access licensed under a Creative Common Attribution 3.0 Generic License. https://creativecommons.org/licenses/by/3.0/legalcode

## Author contributions

C.S. and M.G.G carried out the research and collected and analyzed data. C.S. designed the figures and wrote the manuscript. L.A.H., D.C., and T.M. performed and analyzed lipidomic data and wrote the method for lipidomic. L.A.H., M.R.D., and B.A. critically reviewed the manuscript. X.L. critically reviewed the manuscript. C.K. wrote the manuscript and M.K.A. conceived and designed the study, performed the MRI and MRS experiments, analyzed and interpreted the results, designed the figures, and wrote the manuscript. M.K.A. is the guarantor of this work and, as such, had full access to all the data in the study and takes responsibility for the integrity of the data and the accuracy of the data analysis. All authors approved the final version of the manuscript.

## Funding

## Competing interests

The authors declare no competing interests.
