## [Peer Review File · Communications Biology]

Reviewers' comments:

Reviewer #1 (Remarks to the Author):

The paper by Savva and colleagues utilises a model of maternal obesity to examine sexual dimorphism in whole body metabolism and the hepatic lipidome and potential mechanisms involved in sex-specific outcomes in the setting of developmental programming. Although the data presented add to the field around programming of lipid metabolism and importantly address sex-specific effects which are often overlooked in these types of studies, there are a number of points around the experimental model which require clarification.

The biological replicate number is relatively low for a study of this type i.e. n=6. With an n=6 litters per group how was the n=10 per offspring group derived and was litter effect accounted for? Did all pregnancies deliver litters? Why was the post-weaning housing not standardised (states was 3-5 per cage) as this results in a very low cage number re assessment of food intake and also potentially impacts on food intake and behaviour as hierarchies can develop, particularly in a cage of 5.

Why were GTTs not performed on all animals? i.e. page 7 states 7-10 animals per sex and for other measures not all animals are reported e.g. 6-7 per group.

Can the authors reconcile the lack of difference in offspring body weight in females given a number of reports using similar models in rodents showing clear effects of a maternal obesogenic diet on programming of body weight in offspring.

Why was liver histology not undertaken in the present study as tissue would have been available? – having histological measures on steatosis would be been a valuable addition given the literature around programming of fatty liver disease.

Were the data tested for outliers? Further, some of the differences observed are of statistical significance but of small absolute change thus may be of minor physiological relevance.

The work would benefit from a thorough proofing of English and grammar as there are a number of typographical errors, some of which are detailed below. There are a large number of figure presented – could some of these be consolidate or where no differences exist be simply placed in the text rather than as a figure?

Minor

Page 2, "evidences" should be "evidence"

Page 3, Introduction, 1st para, typo "continous"

Page 5, more details on the diets should be provided e.g. D12451 is 45% kcals from fat, D12450H is 10% kcals from fat

Page 5, What is two by two housing? 6 cages of 2 dams?

Page 6, "using an electrocardiogram.."

Page 7 and elsewhere "homa" is normally "HOMA"

Page 7, "and performed" as detailed previously."

Page 8, "were determined as described previously"

Page 12, typos "idnetified", "correesonding", "psectra"

Page 12, "the the"

Page 13 and 15, should be "ANOVA"

Page 14, a number of typos, "Countreverse:", "signicantly", "appetide", "subcutaneous"

Page 14, "Counterwise" should perhaps be "In contrast to females, ..."

Page 16, typo, "containg"

Page 18 and throughout, "genders" should be "sexes"

Page 21, 1st para, typos "throught altered"

Page 25, typos "dietial" should be dietary, "blunded", "differes"

Page 27, number of typos e.g. "disfunctions", "surpringly", "demonstate"

Page 28, typo "speculatue"

Page 29, "In the our.."should be "In our.."

Page 30, typo "rewieved" and "inprinting"

Supplementary Figure S1 – food intake is not n=10 per group as is meaned per cage so this should be made clear.

Reviewer #2 (Remarks to the Author):

This manuscript examined the effect of maternal high fat diet on the offspring lipidome and lipid-related gene expression. The authors used cutting edge in vivo and ex-vivo analysis tools to disentangle a much complex mechanism. They performed a wonderful analysis of the obtained data. The results suggested sex-dependent hepatic lipidome changes and the influence of maternal diet on the lipidome. This manuscript suggested Maternal environmental factors as a new factor in metabolic dysfunction and obesity. Overall this manuscript presented some novel, original conclusion in a much-needed area and will be of great scientific interest for a broad spectrum of researchers.

Major points:

1) This manuscript has some excellent data from a good experimental setup. This includes the physiological adaptation of maternal diet in the offspring, metabolic adaptation, changes in triglyceride levels, changes in phospholipid levels, and changes in lipid metabolizing enzymes. These data are thoroughly analyzed and presented here. However, it is hard to understand how these data are interconnected? Here the major questions would be whether there are any common traits in these lipidome changes in terms of fatty acyl composition, or biosynthesis pathway?

2) Going through so much lipid data and multiple comparisons make it much harder for the reader to understand the major results from each figure. In my view, a more concise figure would help the reader and the flow of the manuscript better.

3) One of the major issues with the manuscript is the writing style. Many statements are complex and can translate into a completely opposite meaning than the authors intended to convey. Here are some of the examples:

"Although F1 females (F) gained body weight (BW) over time (27 weeks) "- I am not sure what the authors intended to say here. Isn't gaining bodyweight part of the normal growth?

"Overall, the ratio SAT:TF was higher than VAT:TF in both genders. SAT and VAT appeared similar in F irrespective of the maternal diet and time period". The second statement means that SAT and VAT levels are similar in females, which is totally contradictory to the previous statement

"The ratio SAT:VAT was therefore lower in M compared to F in all groups and reduced in HF/C in both sexes at MID but normalized at END term" Again I do not understand the meaning of normalized here?

"Overall, P-PE were significantly different between M and F to 83% (10/12) in C/C and 33% (4/12) in HF/C" – Here it is not clear what the authors are referring to as percentage? May be paraphrasing the statement something similar to "Overall, 83% and 33% of the P-PE species were significantly varied between M and F in C/C and HF/C groups respectively" would be better.

These are only a few examples, the manuscript is filled with such a complex, hard to understand

statements. Overall, simplifying the writing style of the manuscript would increase the readability of the manuscript to a great extent.

4) The text explaining figure 2I (saturation levels in PUFA) mentioned TG- containing 3 and 4 double bonds as PUFA containing TG. TG has three acyl chains therefore, TG containing 3 double bonds may not have any PUFA and 4 double bonds may have only one PUFA. It would be difficult to decide the fatty acid saturation with the overall double bonds.

5) The distance between two of the same group animals in the tSNE is equal or greater than the distance between two groups. The authors may try a discriminant multivariate analysis.

6) In page 14 the authors say "In order to define if the exposure to standard diet after weaning (3 weeks) counteracts the effect of maternal diet in the short or/and long term, we quantified total body fat (TF) at 12 weeks (midterm, MID) and 22 weeks (endterm, END) after weaning (Fig.1A and C)". Here, the effect of the standard diet can be identified only by comparing the midterm and the end-term with the basal value. Therefore, the statement is misleading.

7) What is the rationale of using the ratio to internal standard as the Y-axis for both plasmalogens? While all others were relative to total PL Was this included in the total PL calculations?

8) Figure 6H is not substantiated by proper experimental evidence.

a) There is an opposite pattern of PG and CL observed in the data. For example, there is no difference between C/C F and HF/C F in the most abundant PG(36:2). However, in case of CL(72:7) and CL(72:8), both the control and high-fat diet group females showed significant differences.

b) There are no direct evidence provided to link the FA uptake and de nova synthesis with PG, CL synthesis.

c) Similarly, the previous results on TG are contradicting the notion of lipid droplet increase in Male mice. TG is the major lipid of lipid droplet and there were no differences observed in TG levels in the male mice due to the maternal diet.

9) All the heat map figures lacking quantitative information. Please provide the fold changes corresponding to the colors. The size of the p-value bubble and the symbols are not matching

10) I would suggest the authors deposit the raw MS data on a public repository.

Minor suggestion:

1) 'trimester' instead of 'semester' – introduction

2) Please mention the column used for the LC

3) 'fourth' instead of 'forth' on page 17

4) Please label the axis in Fig 4a

Reviewers' comments:

Reviewer #1 (Remarks to the Author):

The paper by Savva and colleagues utilises a model of maternal obesity to examine sexual dimorphism in whole body metabolism and the hepatic lipidome and potential mechanisms involved in sex-specific outcomes in the setting of developmental programming. Although the data presented add to the field around programming of lipid metabolism and importantly address sex-specific effects which are often overlooked in these types of studies, there are a number of points around the experimental model which require clarification.

The biological replicate number is relatively low for a study of this type i.e. n=6. With an n=6 litters per group how was the n=10 per offspring group derived and was litter effect accounted for? Did all pregnancies deliver litters? Why was the post-weaning housing not standardised (states was 3-5 per cage) as this results in a very low cage number re assessment of food intake and also potentially impacts on food intake and behaviour as hierarchies can develop, particularly in a cage of 5.

Response: In the manuscript, we have now specified the number of animals used in each experimental setting. In brief, we fed twelve F0 dams either a control diet (CD; n=6) or a high fat diet (HFD; n=6) and investigated the metabolic effects on the litter. Each dam delivered litters in different proportions. Dams on CD delivered in average seven to eight littermates (same ratio of female and male offspring) whereas dams on HFD delivered less (five in average). Some of the offspring died at very early age of unknown reasons. The final number of offspring used for this study was 11 females and 12 males born from dams on CD and 11 females and 10 males born from dams on HFD. Each individual was followed over the course of the experimentation for body weight and food intake (six months). However, not all animals were used for every *in vivo* and *ex vivo* experiment justified under consideration of the 3Rs, sample throughput capacity and financial constraints (magnetic resonance, lipidomic, qPCR, and tolerance tests). However, all experimentation was performed according to a prior power calculation and published reports (PMID: 30808418, 23446231, 31811898, 25694038 and 31820027). Throughout the study, we opted for randomized experimental design (random selection of the animals for *in vivo* and *ex vivo* experiments).

After weaning, up to five littermates were housed per cage. However, some male individuals had to be separated due to aggressive behaviors. When females showed hierarchical behaviors in the cage, we separated them to be sure that each individual had full access to food. Since mice are social animals, no individual was housed alone.

Why were GTTs not performed on all animals? i.e. page 7 states 7-10 animals per sex and for other measures not all animals are reported e.g. 6-7 per group.

Response: As described above, according to the laboratory animal welfare regulations, not all animals can be used for all the diverse experiments used in this study without being exposed to too much stress. Since habituation is not always possible (e.g. to MRI and MRS experiments for which animals needed to be anesthetized), we had to reduce the exposure to stress to improve physical and psychological wellbeing of the animal and thereby the experimental outcomes.

Can the authors reconcile the lack of difference in offspring body weight in females given a number of reports using similar models in rodents showing clear effects of a maternal obesogenic diet on programming of body weight in offspring.

Response: The literature is still controversial, possibly due to slightly different experimental settings (e.g. the exact composition of the control and high fat diet, exposure time to the different diet). In the current study we used a match control diet of the high fat diet to minimize potential diet-derived signaling molecule effect. We found several publications in lines with our findings (PMID: 30405201, 31690792 and 29972240). In our study, we assessed the changes in body weight over a long time in the offspring, whereas only few studies show detailed measurements over time, which makes it more difficult to compare across studies.

Why was liver histology not undertaken in the present study as tissue would have been available? – having histological measures on steatosis would be been a valuable addition given the literature around programming of fatty liver disease.

Response: We performed H&E staining on liver cross-sections. We did not observe any histological differences in the livers between the four groups (C/C M and F and HF/C M and F). We have now added these information under the result section: Page 8 “However, liver histology showed no differences in lipid droplets accumulation across all groups (Suppl.Fig.S2a).

It is interesting to note that even though we could not see any clear histological differences between the groups, the lipid species composition was highly sex and maternal diet dependent.

Were the data tested for outliers? Further, some of the differences observed are of statistical significance but of small absolute change thus may be of minor physiological relevance.

Response: We used the whole group for each statistical analysis. Each individual measurement is shown in all graphs (according to the journals requirements). We believe that this clarifies how many individuals were included in each experiment. To take outliers into considerations, we applied a nonparametric K-S test and confirmed the statistical significance of the previously performed parametric T-TEST.

We agree with the reviewer that a small absolute change can be, in some occasion, of minor physiological relevance. However, in some cases it can be relevant as lipid species are signaling molecules which can activate biological pathways and hereby induce significant physiological response.

The work would benefit from a thorough proofing of English and grammar as there are a number of typographical errors, some of which are detailed below.

Response: We have revised and thoroughly proof-read the manuscript. We believe that this has also improved the clarity of our findings.

There are a large number of figure presented – could some of these be consolidate or where no differences exist be simply placed in the text rather than as a figure?

Response: We have tried to consolidate the figures when possible. We used different techniques to assess biological processes related to lipid biosynthesis and metabolism. This

resulted in a large set of independent data from which we selected the most important differences in the F1 offspring dependent on the maternal diet. For completion, we submitted the raw, processed and normalized data to allow assessment of any measurement not mentioned in this manuscript.

<https://figshare.com/s/cc293caa383b439ea0be>

MetaData.txt

Phospholipids_normalized.txt

Phospholipids_raw.txt

Triglycerides_normalized.txt

Triglycerides_raw.txt

Minor

Page 2, “evidences” should be “evidence”

Page 3, Introduction, 1st para, typo “continous”

Page 5, more details on the diets should be provided e.g. D12451 is 45% kcals from fat, D12450H is 10% kcals from fat

Page 5, What is two by two housing? 6 cages of 2 dams?

Page 6, “using an electrocardiogram..”

Page 7 and elsewhere “homa” is normally “HOMA”

Page 7, “and performed” as detailed previously.”

Page 8, “were determined as described previously”

Page 12, typos “idnetified”, “correesonding”, “psectra”

Page 12, “the the”

Page 13 and 15, should be “ANOVA”

Page 14, a number of typos, “Countrewise:”, “signicantly”, “appetide”, “subcutaneous”

Page 14, “Counterwise” should perhaps be “In contrast to females, ...”

Page 16, typo, “containg”

Page 18 and throughout, “genders” should be “sexes”

Page 21, 1st para, typos “through altered”

Page 25, typos “dietial” should be dietary, “blunded”, “differes”

Page 27, number of typos e.g. “disfunctions”, “surpringly”, “demonstate”

Page 28, typo “speculatue”

Page 29, “In the our..” should be “In our..”

Page 30, typo “rewieved” and “inprinting”

Response: We corrected all typos and added further information.

Supplementary Figure S1 – food intake is not n=10 per group as is meant per cage so this should be made clear.

Response: We have changed the supplementary figure legend and clarified how food intake was measured.

Reviewer #2 (Remarks to the Author):

This manuscript examined the effect of maternal high fat diet on the offspring lipidome and lipid-related gene expression. The authors used cutting edge in vivo and ex-vivo analysis tools to disentangle a much complex mechanism. They performed a wonderful analysis of the

obtained data. The results suggested sex-dependent hepatic lipidome changes and the influence of maternal diet on the lipidome. This manuscript suggested Maternal environmental factors as a new factor in metabolic dysfunction and obesity. Overall this manuscript presented some novel, original conclusion in a much-needed area and will be of great scientific interest for a broad spectrum of researchers.

Major points:

1) This manuscript has some excellent data from a good experimental setup. This includes the physiological adaptation of maternal diet in the offspring, metabolic adaptation, changes in triglyceride levels, changes in phospholipid levels, and changes in lipid metabolizing enzymes. These data are thoroughly analyzed and presented here. However, it is hard to understand how these data are interconnected? Here the major questions would be whether there are any common traits in these lipidome changes in terms of fatty acyl composition, or biosynthesis pathway?

Response: we agree with the reviewer that it is not clear yet whether there are any common traits in these lipidome changes in terms of FA composition or biosynthesis pathways. In our study, the F1 offspring received a control diet whereas their mothers were either on control or high fat diet. Since mainly the liver is responsible for lipid synthesis, most lipid species will derive from hepatic *de novo* synthesis.

PLs are crucial to maintain membrane biophysical function with selective organelle-based processes, including membrane-protein trafficking and function (PMID: 25906908). As a consequence, imbalance between SFA, MUFA and PUFA can result in dramatic cellular dysfunctions including metabolic syndrome. We tried to extract the FA composition of PL species as presented in Suppl. Table S2 and S3. However, for some lipid species we were not able to define the FA chain contained into the PL and for others, we defined two to three potential fatty acyl chains, without knowing the exact proportion of each fatty acyl chain, which made it difficult to establish the link between FA composition and PL biosynthesis.

2) Going through so much lipid data and multiple comparisons make it much harder for the reader to understand the major results from each figure. In my view, a more concise figure would help the reader and the flow of the manuscript better.

Response: We agree with the reviewer and have tried to consolidate the figures when possible. We used different techniques to assess biological processes related to lipid biosynthesis and metabolic pathways. This resulted in a large set of independent data from which we selected the most important differences in the F1 offspring dependent on the maternal diet. We further reduced the information presented in the Figures and clarified the text accordingly. For completion, we submitted the raw, processed and normalized data to allow assessment of any measurement not mentioned in this manuscript.

<https://figshare.com/s/cc293caa383b439ea0be>

MetaData.txt

Phospholipids_normalized.txt

Phospholipids_raw.txt

Triglycerides_normalized.txt

Triglycerides_raw.txt

3) One of the major issues with the manuscript is the writing style. Many statements are complex and can translate into a completely opposite meaning than the authors intended to convey.

Here are some of the examples:

“Although F1 females (F) gained body weight (BW) over time (27 weeks) “– I am not sure what the authors intended to say here. Isn’t gaining bodyweight part of the normal growth?
Response: we have changed the sentence to clarify the meaning. Page 5 “Females (F) from week 8 of age until sacrifice weighed less than males (M), irrespective of the maternal diet”

“Overall, the ratio SAT:TF was higher than VAT:TF in both genders. SAT and VAT appeared similar in F irrespective of the maternal diet and time period”. The second statement means that SAT and VAT levels are similar in females, which is totally contradictory to the previous statement

Response: we agree with the reviewer and we have modified the paragraph accordingly. Page 5 “Closer inspection of subcutaneous (SAT) and visceral (VAT) adipose tissue showed sex difference in fat distribution. SAT and VAT appeared unchanged in F irrespective of the maternal diet and time period. However, HF/C M showed reduced amount of SAT at MID and even less at END but gradually more VAT when compared to C/C M and to F groups (Fig.1e-1f)”.

“The ratio SAT:VAT was therefore lower in M compared to F in all groups and reduced in HF/C in both sexes at MID but normalized at END term” Again I do not understand the meaning of normalized here?

Response: we have changed the sentence to clarify the meaning. Page 5 “The ratio SAT:VAT was therefore lower in M compared to F in all diet groups and reduced in HF/C in both sexes at MID but normalized to C/C groups at END (Suppl.Fig.S1c). In sum, our results revealed that maternal diet affects fat distribution in F1 offspring in a sex-specific manner. Interestingly, our results indicate that post-weaning CD of F1 offspring can diminish the effects of maternal HFD on total body fat in the long term (25 weeks) in both sexes.”

“Overall, P-PE were significantly different between M and F to 83% (10/12) in C/C and 33% (4/12) in HF/C” – Here it is not clear what the authors are referring to as percentage? May be paraphrasing the statement something similar to “Overall, 83% and 33% of the P-PE species were significantly varied between M and F in C/C and HF/C groups respectively” would be better.

Response: we agree with the reviewer and we have changed the sentence accordingly. Page 14: “We measured the relative abundance and composition of alkylacyl phospholipids by LC-MS. Of the eight P-PC species detected, we found 20% (1/5) in C/C and 40% (2/5) in HF/C significantly different between M and F (Fig. 5a); and 92% (11/12) and 33% (4/12) of the P-PE species were significantly varied between M and F in C/C and HF/C groups, respectively (Fig. 5b). Overall, relative content and percentage of P-PC (Fig. 5c) and P-PE (Fig. 5d) were similar between sexes in C/C, but were considerably higher in HF/C F than in HF/C M.”

These are only a few examples, the manuscript is filled with such a complex, hard to understand statements. Overall, simplifying the writing style of the manuscript would increase the readability of the manuscript to a great extent.

Response: We addressed the examples highlighted by the reviewer. In addition, we reviewed our statements throughout the manuscript and revised to provide more clarity.

4) The text explaining figure 2I (saturation levels in PUFA) mentioned TG- containing 3 and 4 double bonds as PUFA containing TG. TG has three acyl chains therefore, TG containing 3 double bonds may not have any PUFA and 4 double bonds may have only one PUFA. It would be difficult to decide the fatty acid saturation with the overall double bonds.

Response: we have changed the sentence accordingly. Pages 9/10 “The chemical properties of FA of the TG commonly define their biological relevance. Based on whether double bonds between the hydrogen atoms can be formed, FA are divided into saturated (no double bonds) and unsaturated (with double bonds). Several metabolic diseases are associated with the degree of hydrogen atom bound saturation within the FA of the TG. We, therefore, inspected the level of TG saturation. In C/C group, the relative level of monounsaturated TGs (containing one double bond) was higher in M than in F and, polyunsaturated TGs (containing three and four double bonds) were lower in M compared to F (Fig. 2i). Surprisingly, maternal HFD severely redistributed the abundance of TG containing double bonds in a sex-dependent manner (Fig. 2i).”

5) The distance between two of the same group animals in the tSNE is equal or greater than the distance between two groups. The authors may try a discriminant multivariate analysis.

Response: we thank the reviewer for pointing this out and we have now replaced the tSNE with a principal component analysis. The PCA separated the assessed PL classes into four distinct groups, which clustered HF/C and C/C as well as F and M (new Fig.4a). This indicates that the maternal diet regulates the synthesis and composition of lipid classes differently in both sexes

6) In page 14 the authors say “In order to define if the exposure to standard diet after weaning (3 weeks) counteracts the effect of maternal diet in the short or/and long term, we quantified total body fat (TF) at 12 weeks (midterm, MID) and 22 weeks (endterm, END) after weaning (Fig.1A and C)”. Here, the effect of the standard diet can be identified only by comparing the midterm and the end-term with the basal value. Therefore, the statement is misleading.

Response: we agree with the reviewer and we have changed the sentence accordingly. Page 5 “In order to define the effect of maternal diet on fat content and distribution in the short or/and long term, we quantified total body fat (TF) 12 weeks (midterm, MID) and 22 weeks (endterm, END) after weaning. MRI confirmed that HF/C M but not HF/C F accumulated more fat than C/C animals at MID which was normalized at END (Fig.1d)”

7) What is the rationale of using the ratio to internal standard as the Y-axis for both plasmalogens? While all others were relative to total PL Was this included in the total PL calculations?

Response: we thank the reviewer for pointing this out, we apologize for the mistake and have corrected the Y-axis legend in the plasmalogens graphs.

8) Figure 6H is not substantiated by proper experimental evidence.

Response: In the Figure 6h we summarized the lipidomic data by including sex differences in response to maternal diet (high fat fed *versus* control fed dam). Our data revealed that, when born from obese mothers, F increased plasmalogen species that may facilitate membrane trafficking, including fatty acids, and prioritized synthesis of phosphoglycerides and cardiolipin species, as compared to females born from lean mothers (Fig.2j and Fig.3a). In contrast, M born from obese mothers showed induced level of SFA and very long chain TG as compared to F, and higher level of ceramides (Fig. 5k-5i). We summarized this in a new cartoon, that we modified in Fig.6h.

a) There is an opposite pattern of PG and CL observed in the data. For example, there is no difference between C/C F and HF/C F in the most abundant PG(36:2). However, in case of

CL(72:7) and CL(72:8), both the control and high-fat diet group females showed significant differences.

Response: the main observation with CL species is that all species were significantly increased in the liver of F HF/C compared to F C/C. PG species were unchanged or slightly increased by maternal diet in F and oppositely in M, most of PG species were slightly lowered and two of them significantly down PG(38:4) and PG(38:6) in HF/C compared to C/C. Consequently, F showed higher level of most PG species when born from HF/C compared to M. We clarified these main findings in the Fig.6h

b) There are no direct evidence provided to link the FA uptake and de nova synthesis with PG, CL synthesis.

Response: we agree with the reviewer that there is no direct evidence. However, on control diet (10% fat), most of the FA are synthesized *de novo* by the liver to be released as signaling molecule or as building block of other lipid species including PL. In the Fig6h and 6i we speculate on potential mechanism by which maternal high fat diet will reprogram lipid pathways differently in both sexes.

c) Similarly, the previous results on TG are contradicting the notion of lipid droplet increase in Male mice. TG is the major lipid of lipid droplet and there were no differences observed in TG levels in the male mice due to the maternal diet.

Response: we agree with the reviewer that there are no major differences in TG levels in liver of M compared to F HF/C. Offspring were all fed control diet after weaning and therefore showed very little lipid droplets accumulation in the liver. We believe M HF/C are more affected than F HF/C by maternal obesity due to differences in the transcriptional or posttranscriptional regulation of genes involved in lipid biosynthesis and metabolism.

9) All the heat map figures lacking quantitative information. Please provide the fold changes corresponding to the colors. The size of the p-value bubble and the symbols are not matching

Response: the reviewer is right. We decided not to show fold changes to make it easier for the reader to understand the data. However, we added individual values in bar charts in the supplementary figures if the reader has a particular interest in one lipidomic/gene data. We also fixed the diameter of the circle representing the p-value as pointed out by the reviewer.

10) I would suggest the authors deposit the raw MS data on a public repository.

Response: We have now deposited the raw, processed and normalized data on a public repository. <https://figshare.com/s/cc293caa383b439ea0be>

MetaData.txt

Phospholipids_normalized.txt

Phospholipids_raw.txt

Triglycerides_normalized.txt

Triglycerides_raw.txt

Minor suggestion:

1) 'trimester' instead of 'semester' – introduction

Response: we have changed this accordingly

2) Please mention the column used for the LC

Response: The column used for the LC-MS was AscentisSi column (10 cm × 1 mm, 3 μm, Sigma-Aldrich, Darmstadt, Germany). This information is in the experimental section,

subsection “LC-MS analysis of phospholipids (PL), sphingolipids (SL) and triglycerides (TG)”, in the last line of the page 27.

3) ‘fourth’ instead of ‘forth’ on page 17

Response: we have changed this accordingly

4) Please label the axis in Fig 4a

Response: We have corrected the Fig.4a

REVIEWERS' COMMENTS:

Reviewer #1 (Remarks to the Author):

The comments raised by the reviewer have been satisfactorily addressed. Note some minor typos remain e.g. "sagital" in Supplementary Figure S1 and "double bounds" in main text.

Reviewer #2 (Remarks to the Author):

The authors had addressed many of my concerns. The manuscript now reads better. However, there are still some concerns remaining as discussed below:

1. Still the manuscript is diffused and not focused on a few places. Filtering out and presenting only the essential information would help to improve the manuscript.
2. The color schema in the heatmap need to include quantitative information. For example, "Higher in M" needs to include the fold increase. similarly "Lower in M" should have the fold change information.
3. Figure 5P is missing the p-value information.